# An adaptive sliding mode controller with free-will arbitrary time convergence for three-phase rectifiers in autonomous agricultural vehicles

**Omar Talal Mahmood**[1,2]*, **Wan Zuha Wan Hasan**[1]*, **Norhafiz Azis**[1], **Nor Mohd Haziq Norsahperi**[1], **Hafiz Rahidi Ramli**[1], **Luthffi Idzhar Ismail**[1]

1 Department of Electrical and Electronic Engineering/Faculty of Engineering, Universiti Putra Malaysia, Serdang, Selangor, Malaysia, 2 Department of Electrical Engineering Technology/Technical Engineering College of Mosul, Northern Technical University, Mosul, Iraq

* gs60480@student.upm.edu.my; wanzuha@upm.edu.my

**Data availability statement:** All relevant data are within the manuscript and its Supporting information files.

## Abstract

This study describes a novel adaptive free-will arbitrary time sliding mode controller (AFWATSMC) designed to improve the performance of a three-phase rectifier in an autonomous oil palm grabber vehicle (Robot Autonomous Mechanical Buffalo Grabber (MBG)). The graber, initially powered by a diesel engine with an uncontrolled rectifier, was upgraded to support intelligent systems that require stable DC voltage management. To address the limitations of conventional rectifiers, the suggested AFWATSMC integrates adaptive factors to improve the performance of the original free-will arbitrary time algorithm. The key innovation of this work lies in combining the adaptive sliding-mode control structure with the free-will arbitrary convergence time algorithm, permitting user-defined system settling time nevertheless of dynamic uncertainties (system parameters and initial conditions), a capability not demonstrated in prior rectifier control strategies to the best of the current knowledge. An optimized control laws using genetic algorithm (GA) and particle swarm optimization (PSO) is suggested to tune the parameters using MATLAB Simulink and coding. A smooth waveform with reduced ripple factor was achieved for the DC output of the alternator with a total improvement of 75.47%; the AC output alternator current exhibited an enhanced sinusoidal shape, a reduction of the total harmonic distortion (THD) with a 46.69% improvement, and an achievement of unity power factor of 0.20% improvement was obtained compared to another adaptive SMC.

## Introduction

Given the expanding global demand, the oil palm industry has developed-integrating advanced agronomic techniques, mechanization, and accuracy agriculture to optimize productivity and address sustainability as well as labor challenges [1]. The conventional oil palm harvesting approaches have relied on manual labor, using equipment such as chisels and sickles connected to long poles [2]. These methods demand physical effort and skill, often

**Funding:** The author(s) received no specific funding for this work.

**Competing interests:** The authors have declared that no competing interests exist.

leading to worker fatigue and potential safety risks. Moreover, human duration limits manual harvesting efficiency, affecting the overall productivity in plantations [3].

The above challenges led to the introduction of mechanized tools [4]and semi-automated [5] approaches aimed at enhancing harvesting effectiveness [6]. Devices such as motorized cutters and mechanical harvesters have been designed to reduce the physical effort of workers and increase the speed of operations [7]. The four-wheel grabber is a versatile mechanized vehicle used for harvesting and the collection of fresh fruit bunches (FFBs). It helps to transport the fruit bunches from the fields to collection points, reducing manual labor efforts. Environmental concern is another challenge in this type of plantation [8] where ecological changes highly affect its productivity. Finally, the increased area of the oil palm fields requires a new effective tool that will assist in controlling the harvesting process effectively [9]. These challenges led to the use of autonomous robotics systems in the oil palm plantation industry [10]. Automation will address the labor shortage by reducing the reliance on human workers [11]. Furthermore, robotic integration in oil palm plantations is cost-effective because it requires a high initial investment, and these robots usually reduce the long-term operational costs [10].

The growth of autonomous agricultural machinery has been induced by the integration of advanced control systems, pointing to improving efficiency, precision, and adaptability in various farming operations. Modern literature stresses the pivotal role of control algorithms, sensor tools, and communication systems in the development of autonomous all-terrain vehicles (AATVs) for agriculture [12]. The authors in this work provide an inclusive overview of control algorithms and techniques, sensors, actuators, and communication tools utilized in agricultural AATVs. Their study emphasizes the evolution of control strategies, including adaptive and robust control methods, to adapt to the dynamic and uncertain environments encountered in agricultural settings. In the domain of navigation and path planning, the use of deep learning techniques was found to be promising [13]. The authors discuss the utilization of deep learning in navigation guidance and control systems of autonomous agricultural machinery, demonstrating advancements in path accuracy and knowledge of avoiding obstacles.

Safety concerns are paramount in the implementation of autonomous agricultural equipment as discussed in [14]. The researches conduct a systematic review aiming to the safety aspects of automated agricultural machinery, surveying environmental perception, risk assessment, and human influences. Their work highlights the necessity for comprehensive safety frameworks to ensure the reliable operation of autonomous systems in agriculture [14]. Furthermore, the integration of artificial intelligence (AI) in autonomous agricultural equipment has been investigated to enhance smart harvesting and field monitoring [15]. The author examines AI-based autonomous systems, highlighting their potential in optimizing crop yield and reducing labor costs through real-time data analysis and decision-making.

These studies collectively contribute to the understanding of control systems in autonomous agricultural machinery, offering insights into current advancements and identifying areas for future research and development. Advances in robotics and automation are positively affecting modern oil palm harvesting practices [16]. Improvements like the autonomous drones [17] for observing fruit ripeness and self-driving trucks [18]for transporting harvested bunches are being investigated to reduce labor shortages and enhance operational efficiency. Image processing techniques and deep learning models like YOLO(13), [17, 19] enable automated detection of ripe fruits using high-resolution cameras on drones or ground robots, achieving high precision in classification. Incorporating LiDAR, ultrasonic sensors, and SLAM algorithms, as well as navigation systems allows autonomous machines

to traverse plantation terrains, avoid obstacles, and reach designated trees [18]. These innovations improve productivity, minimize waste, and support the transition toward fully automated oil palm plantations.

One example of a mechanism used in oil palm harvesting is MBG; it has an alternator that is mechanically coupled with the vehicle's diesel engine. The DC voltage used to charge the vehicle's main battery and provide a variety of DC loads generated using a built-in uncontrolled rectifier. After retrofitting the MBG to an autonomous ground vehicle, additional functions like detecting, grabbing, picking, and releasing FFB can be also achieved [20]. This function requires additional devices like the actuator, cameras, and controller with the main board as the GPU.

This study converts the MBG into an autonomous ground vehicle for oil palm harvesting. According to oil palm harvesting and plantations, which often span vast, widely distributed areas, MBGs must operate for extended periods of around 10 hours per day- to effectively cover and service the entire field [21]. This high utilization rate calls for a robust and reliable DC voltage system capable of continuously delivering the required voltage for onboard autonomous equipment, sensors, and computational units. Some typical power supply possibilities for MBGs involve batteries [22], fuel cells [23], solar cells [24], hybrid power systems (alternator with batteries) [25], and alternators integrated diesel engines [26,27]. Alternators are heavy and bulky but can offer a reliable, long-lasting power source [28].

This study will examine the limitations of the DC voltage supply of the four-wheel grabber used in the autonomous oil palm grabber. This grabber's DC voltage is derived from the three-phase alternator equipped with the grabber diesel engine and an uncontrolled rectifier. An inadequate and unregulated DC voltage supply should be upgraded to dependably power advanced autonomous features and support long-term and continuous field operations [29]. Although these technologies mark a significant step forward in labor efficiency and operational coverage, their effectiveness hinges on a robust, stable, and adaptable power system [30]. The use of uncontrolled rectifiers causes voltage fluctuations, inefficient multi-battery charging, and extreme heat development as well as, modeling significant hazards to onboard electronics and battery lifespan. To address the limitations in the DC voltage system of the MBG, an adaptive controller is developed and implemented using a controlled rectifier with sliding mode control and a free-will arbitrary algorithm. By using this algorithm with the controlled rectifier [31], Total Harmonic Distortion (THD) in the AC current can be eliminated using an effective filtering system [32–34], the ripple factor in the DC output is minimized, and power quality is significantly enhanced, thereby boosting total efficiency and alleviating the overloading effect on the alternator by—ultimately solving thermal issues.

This study is the first application of the AFWATSMC algorithm in power electronics systems, particularly for PWM-based three-phase rectifier controllers. To the best of current knowledge, this algorithm has only been explored in theoretical mathematical studies and has never been applied to real-world power systems or power electronic applications. The key novelty of AFWATSMC lies in its unique ability to arbitrarily set the system's convergence time by the user, independently of system parameters, disturbances, and initial conditions—a significant advantage over conventional controllers. This study not only applies AFWATSMC to a nonlinear three-phase rectifier system but also introduces necessary modifications to adapt it to power electronic switching, dynamic load variations, system disturbances (reference voltage variation, parameters fluctuation, and battery load change), and harmonic distortion reduction (THD). To the best of the latest knowledge, this is the first that AFWATSMC parameters have been optimized using GA and PSO, guaranteeing enhanced system performance, fast convergence within the user's desired free-will time, and advanced DC voltage

regulation. These developments establish AFWATSMC as a robust and scalable control strategy for future power electronic applications. The rest of this study will be divided into five parts: part 2 is for the methods and materials used in this research, and part three is for the results and discussion of the AFWATSMC algorithm. Part four presents the conclusions of the results obtained in this work, and then the limitations are presented in the last section.

## Materials and methods

The proposed system was initiated by converting the uncontrolled rectifier equipped with the MBG alternator to an active rectifier. This conversion is achieved by implementing many design modifications, like designing input and output filters. Furthermore, this conversion required replacing the three-phase diode rectifier with an active rectifier circuit using IGBT or MOSFET devices. Finally, a controller (AFWATSMC and DPSMC) was developed to generate the proper gating signals to the IGBT or the MOSFET to confirm pure DC output, unity power factor, less harmonic content, and track the required reference voltage. The output DC voltage of this controlled rectifier is fed to the MBG batteries via a customized BMS. The BMS dynamically adjusts the reference voltage for the AFWATSMC based on the voltage level of the battery slated for charging. It then updates and readjusts the reference voltage for subsequent batteries in sequence. The above explanation is summarized in Fig 1 below.

### Mathematical modeling of three-phase rectifier

Fig 2 represents the circuit diagram for the three-phase rectifier with a resistance and inductance filter (RL) for the input AC side and a capacitor filter (C) for the DC side. The phase rectifier can be represented in the following four equations [35]:

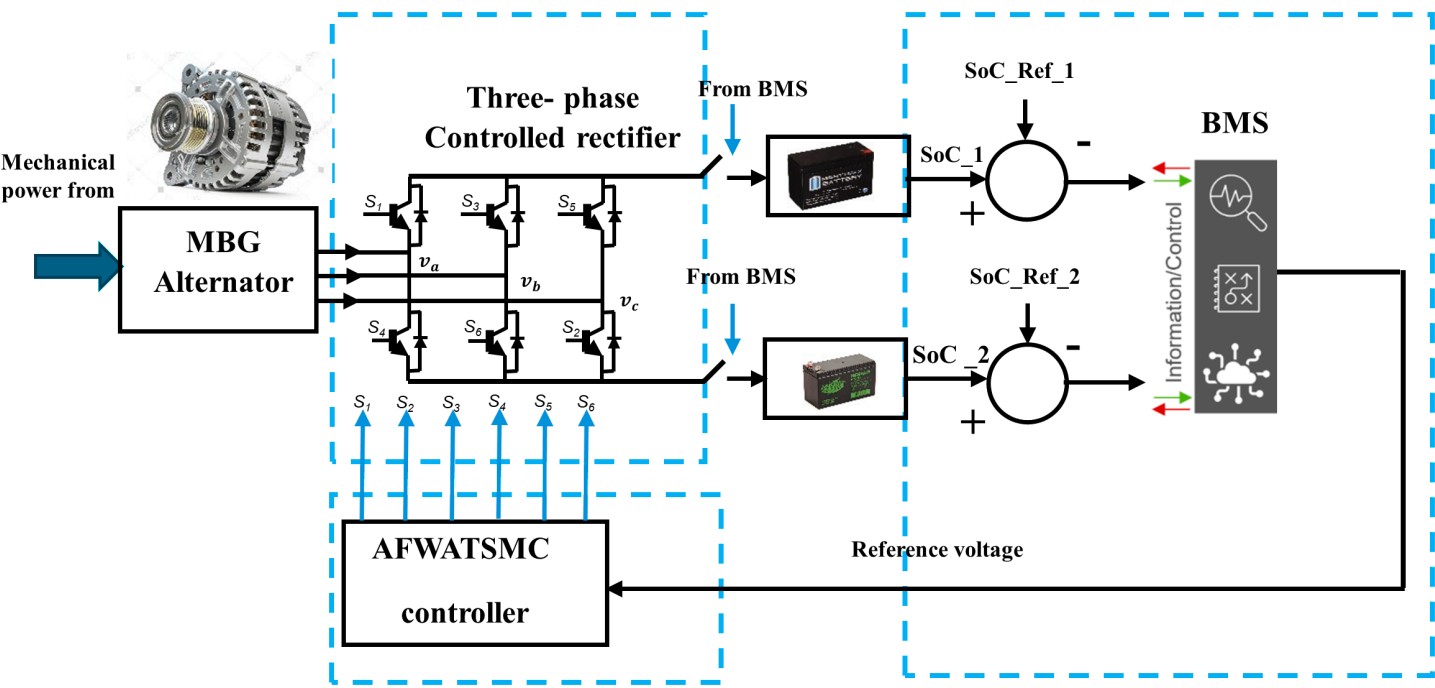

**Fig 1. Research methodology for DC voltage control in the MBG system.**

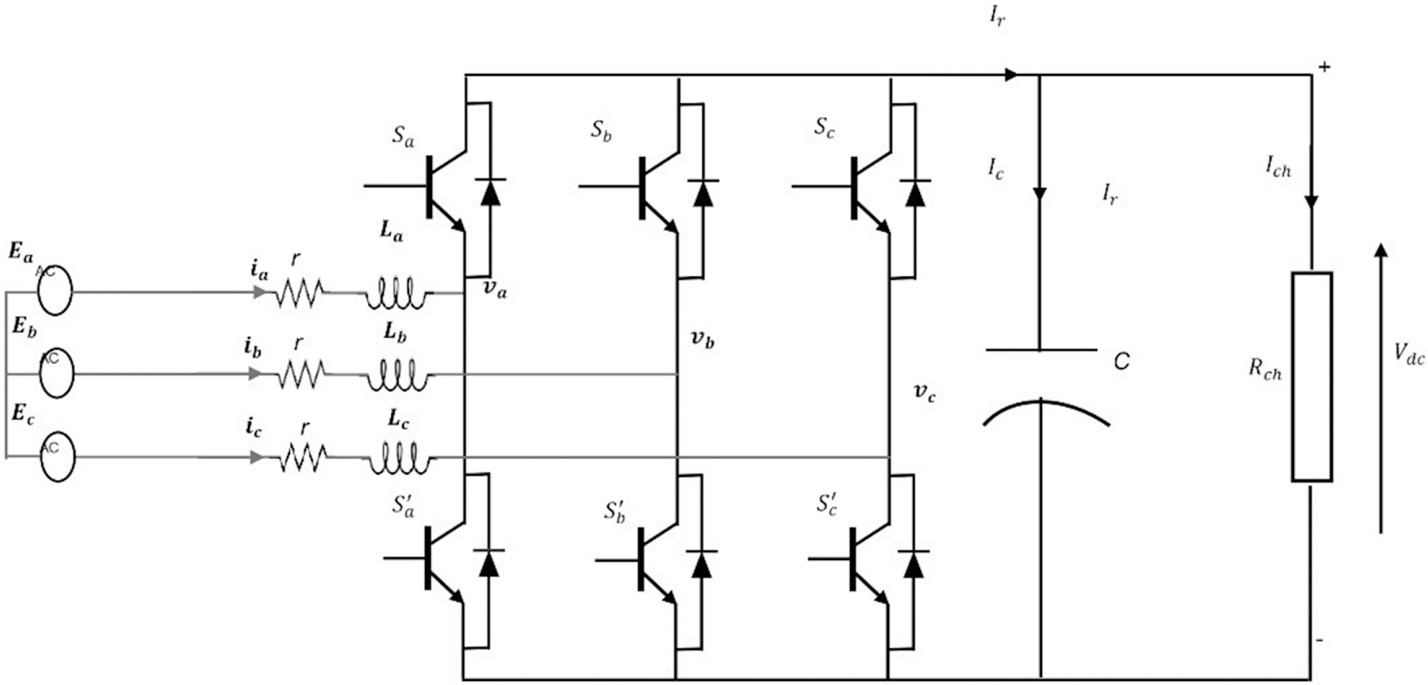

**Fig 2. Detailed schematic of the three-phase PWM controlled rectifier.**

$$L_a \frac{di_a}{dt} = -ri_a + v_a - \frac{V_{dc}}{3}(2S_a - S_b - S_c) \tag{1}$$

$$L_a \frac{di_b}{dt} = -ri_b + v_b - \frac{V_{dc}}{3}(2S_b - S_a - S_c) \tag{2}$$

$$L_a \frac{di_c}{dt} = -ri_c + v_c - \frac{V_{dc}}{3}(-S_a - S_b + 2S_c) \tag{3}$$

$$C\frac{dV_{dc}}{dt} = (S_a i_a + S_b i_b + S_c i_c) - \frac{V_{dc}}{R_{ch}} \tag{4}$$

Where:

- $L_a, L_b, L_c$ and $r$ are the inductance and resistance input filters for each phase.
- $v_a, v_b, v_c$ are the voltages at the AC terminals of the PWM rectifier.
- $i_a, i_b, i_c$ are the phase AC input currents.
- $V_{dc}$ is the DC output voltage.
- $C$ is the DC capacitor filter for the output DC voltage.
- $R_{ch}$ is the load resistance.
- $S_a, S_b, S_c$ are the phase modulation signals for the three phases.

A simplified form of mathematical representation of the three-phase rectifier using dq axes representation as in [35,36] as shown below:

$$L\frac{di_d}{dt} = v_d - Ri_d + \omega L i_q - S_d V_{dc} \tag{5}$$

$$L\frac{di_q}{dt} = v_q - Ri_q - \omega Li_d - S_q V_{dc} \tag{6}$$

$$C\frac{dV_{dc}}{dt} = S_d i_d + S_q i_q - \frac{V_{dc}}{R_{ch}} \tag{7}$$

Where:

- $\omega$ is the angular frequency of the AC-input signal.
- $R$ is the resistance input filter.
- $(S_d)$ and $(S_q)$ are the dq frame modulation signals, and $i_L = \frac{V_{dc}}{R_{ch}}$.

In the inner loop, a direct power SMC controller has been used, so the calculation of the active ($P$) and reactive ($Q$) powers must be achieved in the dq axes as in the following equations:

$$P = v_d i_d + v_q i_q \tag{8}$$

$$Q = v_d i_q - v_q i_d \tag{9}$$

Where ($v_d, v_q, i_d,$ and $i_q$) are the direct and quadrature axes' currents and voltages, respectively. The steps to design the (RL) filters are done using the procedure in [37,38], while the steps to calculate the DC output capacitor filter ($C$) are as in [39,40].

## Design of the three-phase controlled rectifier with voltage and current loop

The controlled rectifier is a boost-type rectifier capable of adapting to the reference change according to the battery's voltage needed to be charged. The outer voltage loop controller used the new adaptive free-will arbitrarily time SMC, and the controller of the inner current loop is the DPSMC. The discussion about the principle of the sliding mode controller will be presented; then, the controller design procedure will be explained in detail. The controller has parameters that need to be tuned; this tuning will be discussed using GA and PSO optimization algorithms.

## Preliminaries about SMC

This method involves constructing a sliding mode surface, and the sliding mode controller is designed for a specific class of nonlinear dynamical systems [41]. The system motion is maintained on the manifold ($S$), defined by sliding mode control, which makes the output track a desired input reference [42].

$$S = \sigma(x, t) = 0 \tag{10}$$

The function ($\sigma$) corresponds to the sliding surface, a line, or a manifold. The control law $u$, which is represented by $S$, can only be obtained by guiding the state variables to the sliding manifold; this control law can be represented as:

$$u = u_n + u_{eq} \tag{11}$$

$u_{eq}$ is the equivalent control vector, which can be found by taking the derivative of Eq (10), $u_n$ which is the vector representing the discontinuous control (the correction factor), provided by:

$$u_n = -k_p \, \text{sgn}(S) \tag{12}$$

$k_p$ is a controlled gain, and the sgn is the function $(\text{sgn}(S))$ is denoted as $\frac{S}{|S|}$, the sliding surface is derived from the general form as follows:

$$S = \sigma = \left( \frac{d}{dt} + k \right)^{r-1} e \tag{13}$$

Where $k$ is a positive parameter that can be picked randomly or by applying a straightforward approach that may result in the right choice, $r$ is the degree relative to the system, and $e$ is the error between the output vector and the intended reference input. According to the SMC design process, motion occurs in two stages: first, it reaches the preselected manifold in state space, called the "reaching phase", and then it slides along the manifold with the appropriate attributes, called the "sliding phase" [43]. When used with a three-phase rectifier, sliding mode control helps the system achieve stable and dependable performance by reducing its sensitivity to parameter changes and outside disturbances [44].

The specifics of the control design can change based on the goals of the control and the rectifier's properties. Practical factors and constraints were also taken into account throughout the implementation phase [45,46]. Implementing SMC for the rectifier must start with system mathematical modeling and selecting the design of the proper sliding surface [47]. An adequate control law must be constructed; then, a switching control strategy must be implemented that compels the system's trajectory to arrive at and remain on the sliding surface. This implementation entails varying the control laws according to the system's orientation concerning the sliding surface, which is the principle of variable structure control. Furthermore, good optimization and tunning for the controller parameters must be done. Finally, simulation and validation for the proposed controller must be done using good software [48].

Furthermore, SMC can have many convergence time types, affecting the performance of the three-phase rectifiers [49]. For three-phase rectifiers, the time convergence can be asymptotic [50], fixed time convergence [49], finite time convergence [51], and other types of convergence algorithms; however, based on the available literature, no applications for the free-will arbitrary control are used with power electronic applications, especially the three-phase rectifiers. The proposed AFWATSMC managed to address the issues of chattering problems complexity in controller design, and, most importantly, it can converge to the desired state in free-will arbitrary time (as the user has requested) regardless of system parameters variation and its initial conditions; this will be explained in details in the subsequent sections.

### The nonlinearities of the three-phase rectifier integrated with BMS supplied from MBG alternator system.

The proposed system consists of a combination of AFWATSMC used to control the performance of the rectifier with the alternator, the BMS, and the load with multiple batteries. This combination generates many uncertainties and disturbances in the system, imposing a nonlinear controller like the SMC. The proposed system is designed to cancel these nonlinearities or disturbances using the advanced adaptive SMC controller. Generally, the control system has two main types of disturbances (matched and unmatched). In summary, SMC is preferred in the system for its robustness against uncertainties and disturbances and its invariance property.

In the current system, there are many matched disturbances, firstly the alternator line voltage variations [52], then load fluctuations in the autonomous MBG [53], the MBG reference

DC voltage changes [54], three-phase parameters [55], initial conditions variations [56], and finally the Harmonics [57] as well as Interharmonics disturbance.

In the power system, an increase in THD will increase the system temperature (machine, transformers, loads, batteries, and other equipment). This increase in temperature will lead to higher temperature performance due to additional losses caused by the harmonics. External losses in the machine core and windings specifically cause harmonic currents; these increases in losses include hysteresis and eddy current losses, which caused a fundamental rise in temperature [58–60]. A commonly used approximation in power systems indicates that every 1% reduction in THD can reduce entire losses by about 2–3%. This loss reduction is immediately strapped to temperature, with a 1% reduction in THD regularly causing about a 0.5–1°C reduction in winding temperature under typical operating circumstances [59,61,62]. Using the adaptation and advanced control algorithm, the proposed controller significantly reduces the THD value.

## Free-will arbitrary time stability algorithm FWAT

A new achievement in nonlinear and linear control theory has been introduced recently: Free-will Arbitrary Time (FWAT) convergence [63] allows the system designer to define the convergence time of the system parameters and states in initial conditions. This ability is metamorphic for applications demanding precise timing constraints, offering accurate control and superior system robustness [64,65]. For instance, the finite-time convergence is associated with system initial conditions, and the convergence time cannot be predefined obviously, limiting the range of its practical applications.

Regarding the fixed time of convergence, the convergence is unrelated to the system's initial conditions but still depends on the system's parameters; thus, it has less flexibility in different applications [64,65]. By integrating free-will arbitrary convergence with the SMC (FWATSMC), the above limitations can be addressed by designing time-dependent scaling functions that permit system designers to identify the convergence time freely. This new approach is remarkably appropriate in power electronic, chaotic, and multi-agent systems, where timing precision is necessary [66,67]. FWATSMC is a robust and suitable controller used in the project to control the output voltage and power of the alternators due to its ability to perform robustly in the presence of uncertainties and disturbances [63,68].

**Mathematical representation for the FWATSMC algorithm.** The system below is a time-varying dynamic:

$$\dot{x} = f(t, x, \vartheta), \quad x(t_o) = x_o \tag{14}$$

Where:

- $x \in R^n$ represents system states.
- $\vartheta \in R^n$ represents system parameters.
- $f: R_+ \times R^n \times R^n \to R^n$ represents a nonlinear function.

For the initial conditions $x(t_o) = x_o$, and for the stabilization problem definitions provided in [63,63,65], the following sliding surface $s_n$ and control law $u(t)$ equations are derived:

For an $n^{th}$ order system, the sliding surface $s_n$ is defined as:

$$s_n = x_n + \sum_{i=1}^{n-1} c_i x_i + \phi(t) \tag{15}$$

Where:

- $c_i > 0$: Coefficients chosen to shape the sliding surface.
- $\phi(t)$: A time-varying term that introduces the arbitrary time convergence capability.

The term $\phi(t)$ is designed as:

$$\phi(t) = \frac{\eta}{(T_f - t)^p} \tag{16}$$

Where:

- $\eta > 0$ is a scaling factor to govern the surface dynamic behavior.
- $p$ is a parameter designed to confirm smooth convergence.

The control law $u_n(t)$ is suggested to guarantee that the sliding surface ($s_n$) goes to zero within the specified convergence time $T_f$. The control law usually combines a discontinuous sliding mode term and a compensatory term for the time-varying dynamics:

$$u_n(t) = -k \operatorname{sign}(s_n) - \dot{\phi}(t) \tag{17}$$

Where:

- $k > 0$: Control gain guaranteeing robustness alongside disturbances.
- $\dot{\phi}(t)$: Time derivative of the time-varying term:

$$\dot{\phi}(t) = -\frac{p\eta}{(T_f - t)^{p+1}} \tag{18}$$

## Mathematical representation of voltage outer loop in three-phase rectifiers

This section presents a mathematical representation of the adaptive sliding mode controller (ASMC), which will be the basis of the new AFWATSMC algorithm. Initially, the error signal $e_{dc}$ is the required state that needs to be zero in a free-will time of convergence. To obtain this error [69], the following equation was used, where $V_{dc_{ref}}$ is the reference DC voltage of the rectifier, and $V_{dc}$ is the measured feedback DC voltage: The error is defined as:

$$e_{dc} = V_{dc_{ref}} - V_{dc} \tag{19}$$

The equation below is for the PI sliding surface $S_{dc}(t)$ used in the proposed algorithm is given by:

$$S_{dc}(t) = K_1 e_{dc}(t) + \int e_{dc}(t)\, dt \tag{20}$$

Where $K_1$ is a positive constant representing the gain of the sliding surface. By rearranging, get:

$$i_{dc}^* = C\dot{V}_{dc} + K_1(V_{dc}^* - V_{dc}) + i_L + K_1(V_{dc}^* - V_{dc})\operatorname{sign}(S_{dc}) \tag{21}$$

In their work [70], the researchers develop an adaptive factor ($\beta$) to improve the robustness of the system, which must satisfy all the condition:

$$\frac{V_{dc}}{R_L} \leq \beta \tag{22}$$

$\beta$ is an unknown positive constant representing ambiguous system properties, disturbances, or uncertainties. Its online estimate ($\hat{\beta}$) evolves as:

$$\dot{\hat{\beta}} = \frac{\eta}{C}|S_{dc}| \tag{23}$$

Where: $\eta > 0$ is the positive learning rate for adaptation.

The control law will be:

$$i_{dc}^* = C\dot{V}_{dc}^* + k_1(V_{dc}^* - V_{dc}) + k_2 S_{dc} + \hat{\beta}\,\mathrm{sgn}(S_{dc}) \tag{24}$$

Furthermore, to reduce the chattering created by the discontinuous term $\mathrm{sgn}(S_{dc})$, the authors in [71] used the saturation function $\mathrm{sat}(S_{dc})$, defined as:

$$\mathrm{sat}(S_{dc}) = \begin{cases} +1, & \text{if } S_{dc} > \gamma \\ -1, & \text{if } S_{dc} < -\gamma \\ \frac{S_{dc}}{\gamma}, & \text{if } |S_{dc}| \leq \gamma \end{cases} \tag{25}$$

Where $\gamma > 0$ is a smoothing factor, selected as 0.01 based on empirical tests. The control law was modified as follows:

$$i_{dc}^* = C\dot{V}_{dc}^* + k_1(V_{dc}^* - V_{dc}) + k_2 S_{dc} + \hat{\beta}\,\mathrm{sat}(S_{dc}) \tag{26}$$

Rearranging, the final control law was:

$$i_{dc}^* = C\dot{V}_{dc}^* + k_2 S_{dc} + \hat{\beta}\,\mathrm{sat}(S_{dc}) \tag{27}$$

## Boundness of the proposed FWAT algorithm

Considering the theorem (1) suggested by the authors in [63] to explain the controller's boundness that combines the SMC with the FWAT algorithm. Yet, when applying the above theorem for the three-phase rectifier system, the boundedness of the proposed FWATSMC was realized. The rectifier system contains time-varying parameters like the input's $RL–C$ values, output filters, and initial conditions for the inductance, capacitance current, and voltage, respectively. Many tests were conducted for different values of parameters and initial conditions to check the system's robustness against these changes. Mathematical Representation: For an $RL–C$ rectifier with an input filter: For an RL-C rectifier with an input filter:

$$V(t,x) = \frac{1}{2}Li^2 + \frac{1}{2}Cv^2 \tag{28}$$

Where $i$ and $v$ are the inductor and capacitor current and voltage, respectively. The time derivative is:

$$\dot{V}(t,x) = Li\dot{i} + Cv\dot{v} \tag{29}$$

This derivative must exponentially decay and the values or the filter parameters and system initial conditions fluctuate (increase or decrease) to test the system convergence. By using the proposed algorithm (AFWATSMC) as a three-phase rectifier controller, variations in filter and system parameters ($L, C, R$) and system initial conditions ($i_L(0), v_C(0)$) influence the Lyapunov function bounds ($\beta_1(x), \beta_2(x)$) and decay rate ($\dot{V}(t)$). To validate the feature

of the convergence regarding of the system parameters, the controller has been applied and increases/decrease the parameters and the initial condition by ±10%, 20%, 30%, and ±50%, validating the convergence behavior.

## Mathematical representation of AFWATSMC for voltage outer loop

When the new FWATSMC is applied to the controller of the three-phase rectifier, a voltage loop has been selected to use this algorithm. After many tests were performed to apply the algorithm, no convergence was achieved in the system states. The limitation of the original FWAT is that it does not have a real-time system or any power electronic application (to the best of available knowledge). Furthermore, the control law in 22 has no approach to confirm convergence within a time designed and required by the user ($T_f$).

In addition, the fixed gain $\beta$ is not adaptive, implying the control may drift for changing initial conditions, system parameters, or disturbances. Moreover, the discontinuous sat ($S_{dc}$), part of the present control law can bring high-frequency oscillations (chattering), specifically when the system functions approach the sliding surface $S_{dc} = 0$.

Finally, the fixed gain $\beta$ is non-adaptive; the controller cannot automatically adjust to varying conditions or disturbances. Thus, a modification has been performed to address the problem of inability to use the FWAT algorithm as it is or as the researchers proposed. An adaptation has been presented to the original algorithm, an adaptive free-will arbitrary sliding mode controller AFWATSMC. To address the three issues above, three modifications have been made to the original control law:

## Modified Free-Will Arbitrary Time (FWAT) convergence term

This term has been added to the original control law in the form of **u**; the value of **u** is as in the expression below to introduce a time-bounded convergence phenomenon to the control law:

$$u = -\eta_1 \frac{\exp(e_{dc}) - 1}{\exp(e_{dc}) \left( T_f^{\text{adjusted}} - t \right)} \tag{30}$$

And the control law is as follows:

$$i_{dc}^* = CV_{dc}^* - u + k_2 S_{dc} + \beta_1 \, \text{sat}(S_{dc}) \tag{31}$$

The advantages of these addition are:

- It will provide a free-will convergence to the error of the DC-link voltage of the rectifier, which is the desired state $\left( V_{dc} \to V_{dc}^* \right)$ within an arbitrary time ( $T_f^{\text{adjusted}}$ ) regardless of initial conditions or disturbances.

- It responds dynamically to error value: in the newly added term, the exponential term $\exp\left( e_{dc} \right) - 1$ used to scale the controller value depending on the value of the error     $e_{dc} = V_{dc}^* - V_{dc}$.

- It improved predictability, that is, engineers can have free-will arbitrary time $T_f^{\text{adjusted}}$ for convergence, permitting beneficial control system tuning and design.

## Time adaptive factor

It is an essential modification to the original control law; this adaptation will modify the value of the desired free-will at an arbitrary time of convergence $\left( T_f \right)$ to an adaptive time of

convergence $\left( T_f^{\text{adjusted}} \right)$ keeping the exact value of the user convergence time as:

$$T_f^{\text{adjusted}} = T_f^{\text{original}} \times \text{adaptive\_Tf\_factor} \tag{32}$$

Where:

$$\text{adaptive\_Tf\_factor} = \begin{cases} 2 & \text{if } |e_{dc}| > \text{some\_threshold} \\ 0.5 & \text{otherwise} \end{cases} \tag{33}$$

$$T_f^{\text{original}} = T_f \times 2 \tag{34}$$

$$T_f^{\text{adjusted}} = T_f^{\text{original}} \times \text{adaptive\_Tf\_factor} \tag{35}$$

## Adaptive value of $\beta_1$

This new adaptation will provide more flexibility to manage bulky initial errors; when the error is high (2), the adaptation factor will also be high; when the error is low, the adaptation factor is low (0.5). These two values have been designed according to many tests using Matlab simulation, and they are suitable for the current application of charging multi-batteries using multiple references for the rectifier output DC voltage.

This adaptation is an essential feature in the proposed charging system that will change the reference DC voltage according to the BMS requirements to charge the batteries. Another vital advantage of this adaptation is that it can control the value of the ripple in the DC-link of the rectifier by controlling the value of the adaptive factor.

The value of $\beta$ that is used in the original control law has the below adaptation:

$$\beta_1 = \beta \cdot \max\left( 0, 1 - \frac{|S_{dc}|}{\text{expected\_sd\_max}} \right) \tag{36}$$

This arrangement will adjust the original controller's robustness based on the sliding surface value. This adaptation will improve the robustness of the new controller using the value of $\beta_1$ which will compensate for the system variations in parameter values (uncertainties) and any other external disturbances (like the load resistance variation for different batteries in this system) by scaling the controller dynamically.

Furthermore, this adaptation will also provide a smooth controller effort when the value of $|S_{dc}|$, is large, this will avoid aggressive control behavior. When $|S_{dc}|$ decreased, $\beta_1$ increase, guaranteeing precise tracking at the required state. Finally, this adaptation gave more reduction in the chattering effect; the combination of adaptive $\beta_1$ and the sat $(S_{dc})$, minimizes the chattering effect that can arise in the original law. This adaptation is also crucial to the proposed charging system for the MBG where the error is high, and thus, the value of $|S_{dc}|$, will be high also.

All in all, the above procedure was activated and repeated its cycle whenever the error of the DC-link voltage is over 0.5, which is done using the (persistent) function in Matlab m. file programming [72].

## Dynamic behavior for the AFWATSMC

In this section, the dynamic behavior of the new approach was discussed, and then a variation of the convergence of this approach was considered using the Lyapunov method. There

were three phases of operation for the AFWATSMC in the three-phase rectifier voltage loop: phase 1, which was before transition; phase 2, which was during the transition and finally, phase three after transition, as in the formulas below: **Phase 1, before the transition:** During the time interval $(t < T_f^{\text{adjusted}} - \text{transition\_duration})$, the behavior of $u(t)$ is given by:

$$u(t) = -\eta_1 \frac{\exp(e_{dc}) - 1}{\exp\left(e_{dc}(T_f^{\text{adjusted}} - t)\right)} \tag{37}$$

This adjustment indicates that if the DC-link error ($e_{dc}$) continues, the controller takes aggressive compensation action until the final time. Physically, this controller drives the system to the desired state ($e_{dc} = 0$) before the scheduled end of the adaptation. Regarding the DC voltage loop controller, in this phase (before transition), the voltage reference controller ($i_{dc}^*$) was significantly reduced by $u$. As $u$ raises in magnitude (negatively), it drags ($i_{dc}^*$) downward, pushing the rectifier output voltage to more aggressively fix any lingering DC-link voltage error.

**Phase 2, during transition:** During this phase, the added term $u$ ($T_f^{\text{adjusted}} - \text{transition\_}$ $\text{duration} \le t < T_f^{\text{adjusted}}$), is given by:

$$u(t) = \left(-\eta_1 \frac{\exp(e_{dc}) - 1}{\exp\left(e_{dc}(T_f^{\text{adjusted}} - t)\right)}\right) \times \frac{T_f^{\text{adjusted}} - t}{\text{transition\_duration}} \tag{38}$$

Then , the $(T_f^{\text{adjusted}} - t)$ expressions cancel out, leaving:

$$u(t) = -\eta_1 \frac{\exp(e_{dc}) - 1}{\exp(e_{dc}) \cdot \text{transition\_duration}} \tag{39}$$

During the entire transition period, the controller term $u$ will be a constant value and a function of $\exp(e_{dc})$ and the parameter $\eta_1$, but not a function of time. This guarantees continuity at the beginning of the transition prevents the infinite increment of $u$, rapidly reducing $u$ to zero. Physically, during the transition, the DC-link voltage loop controller exhibits a constant additional negative offset represented by a constant value of $u$. This constant compensation attempts to reduce the error in the DC voltage and stabilize the DC output of the rectifier at the reference voltage within the user's free-will time $T_f$. Consequently, the system is stable, waiting for the next phase to remove the effect of $u$.

**Phase 3, after transition:** In this time interval ($t \ge T_f^{\text{adjusted}}$), the value of the AFWATSMC term will be zero:

$$u(t) = 0$$

At the end of this phase, the proposed controller will be completely switched off. This phase represents the termination of the controller's adaptive feedforward-like behavior.

## Lyapunov validation for the AFWATSMC for the outer voltage loop

The Lyapunov function is used for stability analysis to validate the convergence and stability of nonlinear systems mathematically [73]. Let $V$ be the Lyapunov function. For the proposed system, the Lyapunov candidate equation was assumed as:

$$V = \frac{1}{2}S_{dc}^2 \tag{40}$$

This candidate function is permanently nonnegative, and $V = 0$ only when $S_{dc} = 0$. As $S_{dc}$ is formed from the error $e_{dc}$, then if $S_{dc} \to 0$, it follows that $e_{dc} \to 0$ (nearly zero steady-state error), and its integral does not increase. The derivative of Eq (40) with respect to time was:

$$\dot{V} = S_{dc}\dot{S}_{dc} \tag{41}$$

The derivative of the sliding surface with respect to time was:

$$\dot{S}_{dc}(t) = K_1\dot{e}_{dc}(t) + e_{dc}(t) \tag{42}$$

From the dynamics of the three-phase rectifier DC-link voltage, to have:

$$C\dot{V}_{dc}(t) = i_{dc}^*(t) - i_L(t) \tag{43}$$

Since $e_{dc}(t) = V_{dc,ref} - V_{dc}(t)$, and $V_{dc,ref}$ is typically constant, to get:

$$\dot{e}_{dc}(t) = -\dot{V}_{dc}(t) \tag{44}$$

Without loss of generality, this can be expressed as:

$$C\dot{V}_{dc}(t) = i_{dc}^*(t) - i_{dc,load}(t) \tag{45}$$

Substituting the value of $i_{dc}^*$ into the voltage dynamic equation, to get the following:

$$C\dot{V}_{dc}(t) = C\dot{V}_{dc}^* - u + k_2 S_{dc} + \beta_1 \text{sat}(S_{dc}) - i_L(t) \tag{46}$$

Thus:

$$\dot{e}_{dc}(t) = -\dot{V}_{dc}(t) = -\dot{V}_{dc}^* + \frac{u}{C} - \frac{k_2}{C}S_{dc} - \frac{\beta_1}{C}\text{sat}(S_{dc}) + \frac{i_L(t)}{C} \tag{47}$$

Substituting the expression of $\dot{e}_{dc}(t)$ into $\dot{S}_{dc}(t)$, to obtain:

$$\dot{S}_{dc}(t) = K_1\dot{e}_{dc}(t) + e_{dc}(t) \tag{48}$$

The key definition of Lyapunov-based stability analysis guarantees that the derivative of the Lyapunov candidate function ($\dot{V}$) is always negative:

$$\dot{V} = S_{dc}\dot{S}_{dc} \leq 0 \tag{49}$$

This term $u$ plays an essential role in changing the sign of $S_{dc}\dot{S}_{dc}$ according to the sign of $e_{dc}$ (and accordingly $S_{dc}$), and it increases in magnitude as $t$ reaches $T_f^{\text{adjusted}}$. Thus:

- When the error $e_{dc} > 0$, then the exponential term $\exp(e_{dc}) - 1 > 0$, forcing $u$ to be negative. As $u$ was negative, $\dot{V}_{dc}$ reduces, thus adjusting $\dot{e}_{dc}$ so that $\dot{S}_{dc}$ becomes negative when $S_{dc} > 0$ .This forced the system towards $S_{dc} = 0$.

- When $e_{dc} < 0$, then the term $\exp(e_{dc}) - 1 < 0$, forcing $u$ to be positive. This positive value of $u$ in this approach also guarantees that if $S_{dc} < 0$, $\dot{S}_{dc}$ is positive, pushing $S_{dc}$ back toward zero.

Since the term $\exp(e_{dc}) - 1$ changes sign strictly at $e_{dc} = 0$ and was scaled by the term $(T_f^{\text{adjusted}} - t)$ in the denominator, the controller gains aggressiveness as the final time $T_f^{\text{adjusted}}$ approaches. This arrangement guaranteed that regardless of the sign of $e_{dc}$, the control law forces $S_{dc}$ towards zero by or before the desired final time specified by the user.

## Algorithm for the AFWATSMC

Fig 3 below shows the flowchart for the code implemented in the AFWATSMC algorithm. Below are the steps used to develop the proposed AFWATSMC algorithm flowchart:

a. **Initialization:** The step begins with initializing the algorithm, where key input parameters are fed into the system, such as the reference DC voltage ($V_{dc-ref}$) and the measured DC voltage ($V_{dc}$).

b. **DC-Voltage Error calculations:** This mathematic stage for estimating the error ($e_{dc}$) between the reference DC voltage and the measured DC voltage.

c. **Error Threshold check:** Next, the algorithm evaluate whether the error ($e_{dc}$) exceeds the predefined threshold set by the algorithm. If the error is high, indicating a significant deviation from the required voltage, the algorithm automatically sets the adaptive factor (adaptive_Tf_factor) equal to 2, slowing the convergence process to prevent aggressive control behavior. Conversely, if the error is within an acceptable range, the factor is set to 0.5, allowing for faster convergence.

d. **Free-will Arbitrary Convergence Time Adjustment:** The algorithm dynamically adjusts the convergence time ($T_f$) by using the adaptive factor.

e. **Sliding Surface Design:** The sliding surface ($S_{dc}$) is evaluated by applying errors and their integration.

f. **AFWAT Control Law Application:** The AFWAT term of the control input ($u$) was calculated using the adapted FWAT control law in Eq (31). This control law confirms that the system error converges to zero within an arbitrary time value ($T_f$).

g. **Dynamic Adaptation of $\beta_1$:** The adaptive gain ($\beta_1$) was modified dynamically based on the sliding surface value and subsequently subjected to a saturation function to inhibit excessive control actions.

h. **Direct Axes Current Reference Calculation:** The algorithm determines the reference direct current ($i_{dc}^*$).

i. **Disturbance Monitoring:** The final block is a decision-making step that continuously checks for system disturbances.

## Mathematical representation for the current or inner loop of control

Combining DPC and SMC into three-phase rectifiers provides significant advantages such as rapid dynamic response, robustness, and THD reduction. Contrasting the traditional control algorithms like PI control, the DPC directly controls the active and reactive powers without involving the inner current loop, indicating minimal control effects and fast response dynamics. SMC enhances the system's robustness, ensuring stability and convergence in finite time under system uncertainties and disturbances [35,74].

The sliding surface is constructed to maintain the DC-link voltage close to the reference voltage using the reference current generated from the voltage loop. In addition, the sliding

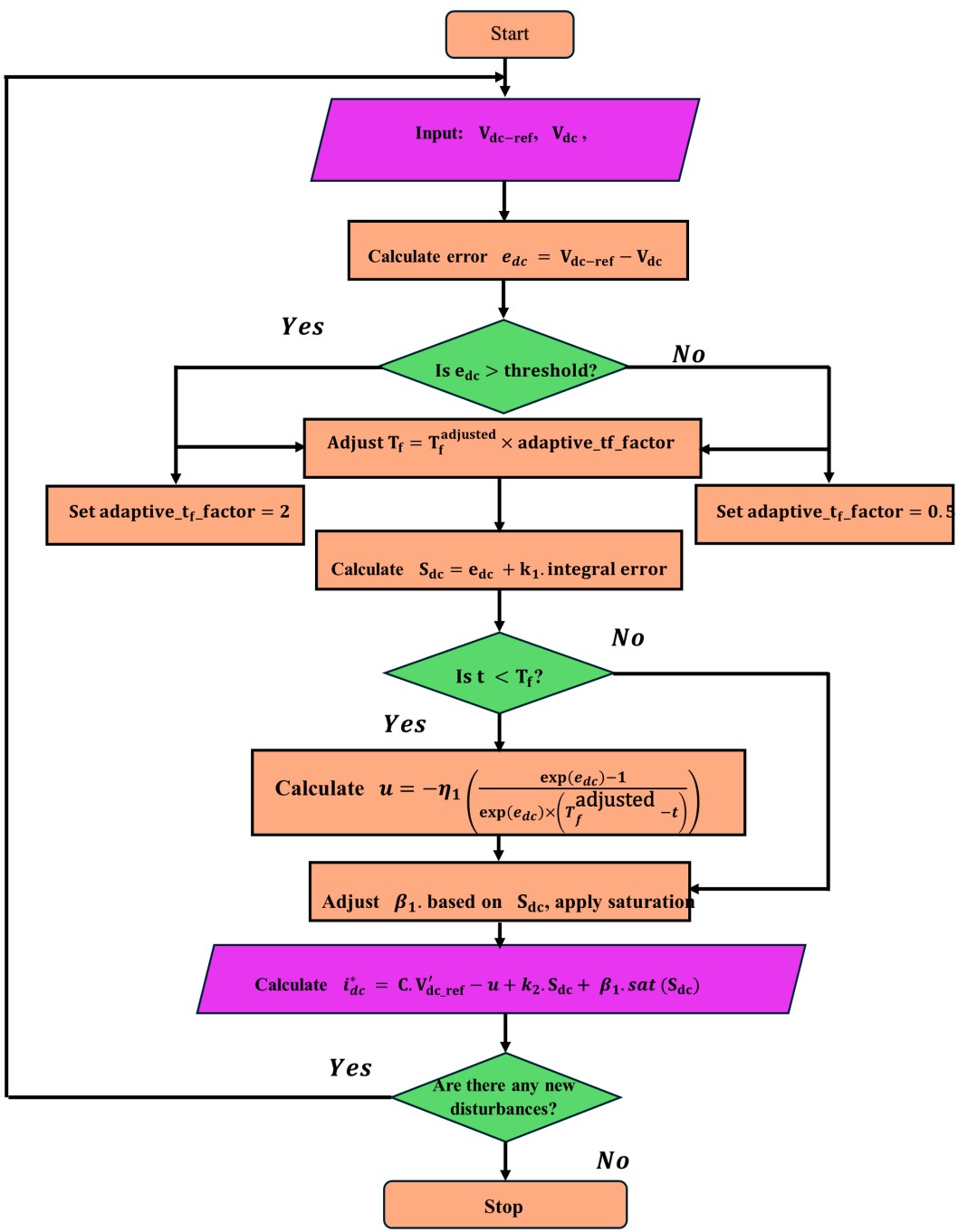

**Fig 3. Flowchart of the proposed Adaptive Free-Will Arbitrary Time Sliding Mode Controller (AFWATSMC) used in the voltage loop of the rectifier controller.**

surface applies to the unity power factor and minimizes the total harmonic distortion by eliminating the value of the reactive power. The methodology of using the DPC-SMC leverages the advantages of the SMC, which are the parameters insensitivity feature, disturbance rejection, and voltage stability in PWM three-phase rectifiers [75,76].

This loop is responsible for the elimination of the error ($e_d$) between the direct current reference ($i_{dc}^*$) and the measured direct current ($i_d$) to ensure reduced steady-state error for the DC input voltage. Additionally, this loop eliminates the error ($e_q$) between the quadrature current reference (which is zero in this project) and the measured quadrature current ($i_q$). This error elimination ensures unity power factor, pure sinusoidal input current, and elimination of THD.

In this approach, the reference active power ($p_{ref}$), calculated from the reference direct axes current ($i_{dc}^*$), will be compared to the measured active power ($p$) to calculate the error ($e_p$). Similarly, the reference reactive power ($q_{ref}$) (zero to maintain unity power factor) was compared to the measured reactive power ($q$). The steps to design the controller for this loop are as follows:

**Selection of the sliding surfaces:**

$$s_p = e_p + k_{ip} \int e_p \, dt, \quad s_q = e_q + k_{iq} \int e_q \, dt \tag{50}$$

Where ($e_p$) and ($e_q$) represent errors in active and reactive powers, respectively. ($k_{ip}$) and ($k_{iq}$) are the control gains in the active and reactive power sliding surfaces.

**Control law design:**

$$U_{dq} = -D_{dq}^{-1} \left[ F_{dq} + \gamma_1 s_{pq} + \gamma_2 \, \text{sgn}(s_{pq}) \right] \tag{51}$$

Where:

$$D_{dq} = \begin{bmatrix} -\frac{e_d}{L} & -\frac{e_q}{L} \\ -\frac{e_q}{L} & \frac{e_d}{L} \end{bmatrix} \tag{52}$$

And:

$$F_d = \dot{e}_d \cdot i_d + \dot{e}_q \cdot i_q + \left( k_{ip} - \frac{R}{L} \right) \cdot p + \frac{1}{L} \cdot (e_d^2 + e_q^2) - p_{ref} - k_{ip} \cdot p_{ref} \tag{53}$$

$$F_q = \dot{e}_d \cdot i_q - \dot{e}_q \cdot i_d + \left( k_{iq} - \frac{R}{L} \right) \cdot q - q_{ref} - k_{iq} \cdot q_{ref} \tag{54}$$

The SMC expression below is used to eliminate the disturbance of THD and ensure the convergence of the system with the required free-will time and unity power factor.

$$\dot{s}_{pq} = -\gamma_1 s_{pq} - \gamma_2 \, \text{sgn}(s_{pq}) \tag{55}$$

Lyapunov stability was utilized again to analyze the stability of this loop. By considering the following Lyapunov function candidate:

$$V = \frac{1}{2} s_{pq}^T s_{pq} \tag{56}$$

The derivative of $V$ is given by:

$$\dot{V} = s_{pq}^T \dot{s}_{pq} \tag{57}$$

By substituting Eqs (55) into (57), it becomes:

$$\dot{V} = -\gamma_1 s_{pq}^T s_p - \gamma_2 s_{pq}^T \text{sgn}(s_{pq}) \leq 0 \tag{58}$$

According to Eq (58), the time derivative of the Lyapunov function $\dot{V}$ if the negative is definite, then the system becomes stable.

## SMC parameters optimization using genetic (GA) and Particle Swarm Optimization PSO algorithms

GA and PSO algorithms were used to tune the controller parameters for the voltage loop controller (AFWATSMC) and DPSMC for the current loop controller. The single-objective GA and PSO algorithms tune $k_1, k_2, \eta_1$ as follows:

$$f(\mathbf{x}) = \sum_{t=0}^{T} \left( V_{dc_{ref}}(t) - V_{dc}(\mathbf{x}, t) \right)^2 \tag{59}$$

Where:
- $\mathbf{x}$ is the AFWATSMC parameter set to be optimized, defined as:

$$\mathbf{x} = [k_1, k_2, \eta_1]$$

$T$ is the total evaluation time for the error. GA is an optimization heuristic emulating the natural selection procedure [77]. To the best of current knowledge, this is the first time AFWATSMC has been integrated with a single-objective GA (SOGA) for parameter tuning. Similarly, PSO was applied to enhance the parameters of the PID controller to improve the performance of the hybrid energy system and enhance efficiency under disturbance influences [78]. Integrating these algorithms with SMC achieves higher performance and robustness in complex nonlinear applications [79].

Additionally, Multiobjective PSO was used to tune the control parameters $G1_P, G2_P, G1_Q, G2_Q$ as follows:

$$f(G1_P, G2_P, G1_Q, G2_Q) = \text{Weight}_1 \cdot |\text{error}_{id}| + \text{Weight}_2 \cdot |\text{error}_{iq}| \tag{60}$$

Where $\text{Weight}_1$ and $\text{Weight}_2$ represent inertia weights controlling the impact of the particle's previous velocity on its current one, providing a balance between the swarm's global and local exploration capabilities. $\text{error}_{id}$ and $\text{error}_{iq}$ are the errors for active and reactive powers, respectively. However, after several iterations, no acceptable results were obtained, indicating that these parameters did not significantly affect the error values. Consequently, the parameters were changed to $(G1_d, G2_d)$ for the direct axes and $(G1_q, G2_q)$ for the quadrature axes. Despite multiple iterations, acceptable results were not achieved (these negative results will be discussed in the results section).

Therefore, Particle Swarm Optimization (PSO) is applied to optimize the parameters of the voltage and current loops of the AFWATSMC and DPSMC, respectively. The adaptive features of this algorithm, integrated with its capability to deal with discrete, continuous, and mixed variables, make it highly relevant to dynamic systems where the control parameters interact in complex ways [80]. Fig 4 below shows the complete representation of the proposed controllers of a three-phase rectifier using Matlab Simulink and coding of m-file.

## Validation of the proposed AFWATSMC algorithm

In this comparison, the model and system proposed in [35,70,71,81], which is the ASMC approach, have been used to compare the performance of the new algorithm. This comparison was made regarding the shape and THD content of the AC output voltage and current of

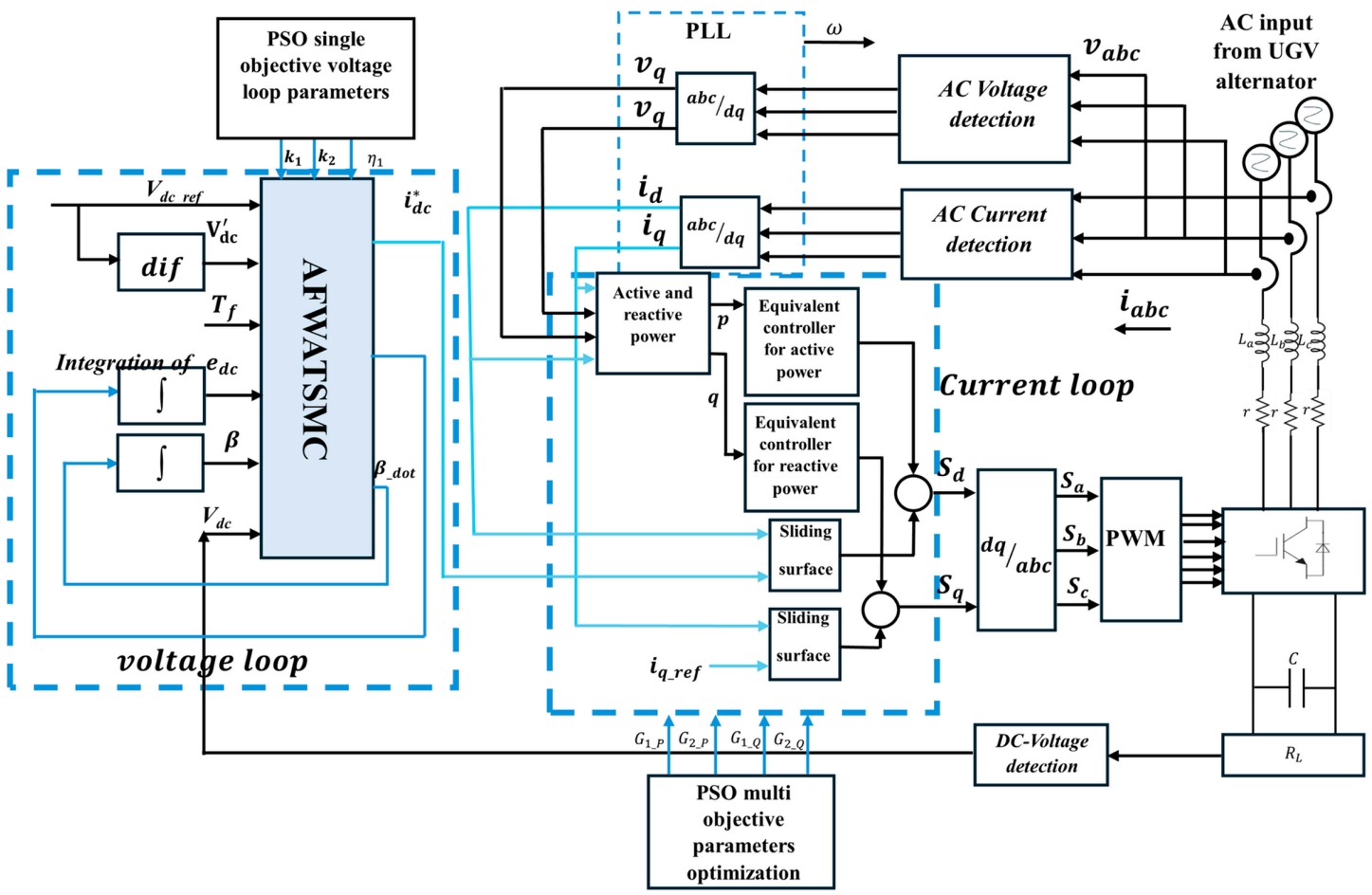

**Fig 4. Schematic Diagram for the Proposed AFWATSMC and DPSMC Controllers used to control the Three-Phase Rectifier in voltage and current loops respectively.**

the alternator. Furthermore, the DC output response is another parameter to compare with its ripple factor. In addition, the time convergence of the system to zero was also compared. The value of the power factor was also compared using the two algorithms. Moreover, the step response parameters are also used as comparison parameters between the two algorithms.

The effect of the THD on the alternator temperature was also compared using the proposed algorithm with the ASMC. The same control law and PI sliding surface have been implemented in the proposed algorithm without adding the ($u$) term in the AFWATSMC algorithm. The same DPSMC is used in the current loop, and optimized SMC parameters are used for both systems. The same AC and DC filter parameters have also been used for both systems.

Finally, the same 15 V as a reference voltage and loading conditions were applied to both systems. Arbitrary time of convergence ($T_f$) of the new algorithm was set to be 0.2 seconds. The mathematical representation for the ASMC was used in Eq (27) above.

In Table 1 below is a summary of the mathematical representation of the new AFWATSMC algorithm compared to the adaptive sliding mode controller proposed in [35,70,71,81].

**Table 1. Mathematical representation of the proposed AFWATSMC compared with ASMC.**

| Item Name | Mathematical representation for the ASMC | Mathematical representation for the AFWATSMC | Advantages |
|---|---|---|---|
| Sliding surface | $S_{dc}(t) = K_1 e_{dc}(t) + \int e_{dc}(t)dt$ | $S_{dc}(t) = K_1 e_{dc}(t) + \int e_{dc}(t)dt$ | Same sliding surface has been used |
| Control law | $i_{dc}^* = CV_{dc}^* + k_2 S_{dc} + \hat{\beta}\,\mathrm{sat}(S_{dc})$ | $i_{dc}^* = CV_{dc}^* + u + k_2 S_{dc} + \beta_1\,\mathrm{sat}(S_{dc})$ | The new term ($u$) guarantees free-will convergence and improves performance. |
| Free-will arbitrary time term | No free-will arbitrary time term. | $u = -\eta_1 \dfrac{\exp(e_{dc}) - 1}{\exp(e_{dc})\left(T_f^{adjusted} - t\right)}$ | - Free-will convergence to DC-link voltage error.<br>- Responds dynamically to error.<br>- Improves predictability. |
| Adaptive value of $\beta_1$ | $\dot{\hat{\beta}} = \dfrac{\eta}{C}\lvert S_{dc}\rvert$ | $\beta_1 = \beta$ <br> $\max\left(0.1,\ 1 - \dfrac{\lvert S_{dc}\rvert}{\text{expected\_sd\_max}}\right)$ | · - Adapts to uncertainties and parameter variations.<br>- Smooth control when $\lvert S_{dc}\rvert$ is large.<br>- Increases $\beta_1$ for precision at low error.<br>- Minimizes chattering with $\beta_1$ and $\mathrm{sat}(S_{dc})$.<br>- Critical for BMS charging where error is high. |
| Time adaptive factor | No time adaptive factor | $T_f^{adjusted} = T_f^{original} \times \text{adaptive\_Tf\_factor}$ <br> $\text{adaptive\_Tf\_factor} =$ <br> $\begin{cases} 2 & \text{if } \lvert e_{dc}\rvert > \text{threshold} \\ 0.5 & \text{otherwise} \end{cases}$ <br> or $T_f^{adjusted} = T_f \times 2$ | - Adapts to initial error magnitude.<br>- High error → large adaptation factor.<br>- Supports SoC-based voltage control in BMS. |
| Lyapunov validation | $V = \dfrac{1}{2}S_{dc}^2$ | $V(t,x) = \dfrac{1}{2}Li^2 + \dfrac{1}{2}Cv^2,$ <br> $\dot{V}(t,x) = Li\dot{i} + Cv\dot{v}$ | -The term $\exp(e_{dc}) - 1$ in $u$ vanishes at $e_{dc} = 0$.<br>- Denominator $\left(T_f^{adjusted} - t\right)$ ensures free-will time convergence.<br>- Ensures $S_{dc}$ reaches zero by target time. |

*Note.* This table compares the adaptive free-will arbitrary time sliding mode controller (AFWATSMC) and the adaptive SMC (ASMC), highlighting improvements in convergence, adaptability, and robustness.

## Summary of tested scenarios and controller performance

To comprehensively evaluate the robustness and adaptability of the proposed Adaptive Free-Will Arbitrary Time Sliding Mode Controller (AFWATSMC), various operating conditions and disturbances were tested. These include parameter variations, reference voltage changes, MBG speed fluctuations, and step responses. For comparative assessment, the ASMC algorithm was also applied under the same scenarios. Table 2 provides a consolidated summary of all tested conditions, the controllers used, the resulting system responses, and the key performance metrics. This overview facilitates a clearer understanding of the controller's capabilities and improvements over conventional methods.

## Results

This section exhibits and analyzes the results acquired from implementing the proposed Adaptive Free-Will Arbitrary Time Sliding Mode Controller (AFWATSMC) compared to the conventional Adaptive Sliding Mode Controller (ASMC) and other tests regarding the disturbance rejection. A series of simulation scenarios were conducted to evaluate the performance of both controllers under different conditions, including parameter variations, reference voltage changes, load disturbances, and system uncertainties. Key performance indicators such as convergence time, ripple suppression, voltage regulation accuracy, total harmonic distortion (THD), and power factor improvement are discussed. The findings are envisioned through

**Table 2. Summary of All tested Conditions and system Responses for controllers evaluation.**

| Tested Condition | Description | Controller Used | System Response | Performance Metrics |
|---|---|---|---|---|
| Parameter variation | L, C, R ±30% | AFWATSMC/ASMC | Stability, convergence under uncertainties | Settling time, overshoot, THD |
| Initial conditions variation | **Capacitor initial voltage** ($v_C(0)$)**, Inductor initial current** ($i_L(0)$) **, ±30% to ±120%** | AFWATSMC | Stability, convergence under uncertainties | free-will convergence time |
| Load change | Battery charging from: 12V to 24V battery | AFWATSMC and ASMC | Voltage tracking and ripple suppression | Ripple %, convergence time |
| Reference voltage change | DC ref: 15V → 28V | AFWATSMC vs ASMC | Fast voltage regulation | Convergence time, tracking error |
| MBG speed disturbance | Alternator speed variation | AFWATSMC | DC link voltage stabilization | Convergence time |
| THD comparison | AC current waveform | AFWATSMC vs ASMC | Sinusoidal waveform with reduced THD | THD % reduction |
| Step response | DC voltage step test | AFWATSMC vs ASMC | Transient response tracking | Overshoot, settling time, rise time |

*Note.* This table summarizes different validation scenarios tested with both AFWATSMC and ASMC controllers.

detailed time-domain response plots, control signal tracking, sliding surface behavior, and power quality metrics to demonstrate the proposed control scheme's robustness, adaptability, and superiority.- MBG Alternator Speed Variation and Load Variation Disturbances.

## MBG alternator speed variation and load variation disturbances

Fig 5 below shows the impact of speed variations on the DC output voltage for the controlled rectifier using the proposed AFWATSMC. The experimentally measured speed for the diesel engine of the experimental MBG is 104.66 rad/sec. In this figure, the speed is varied according

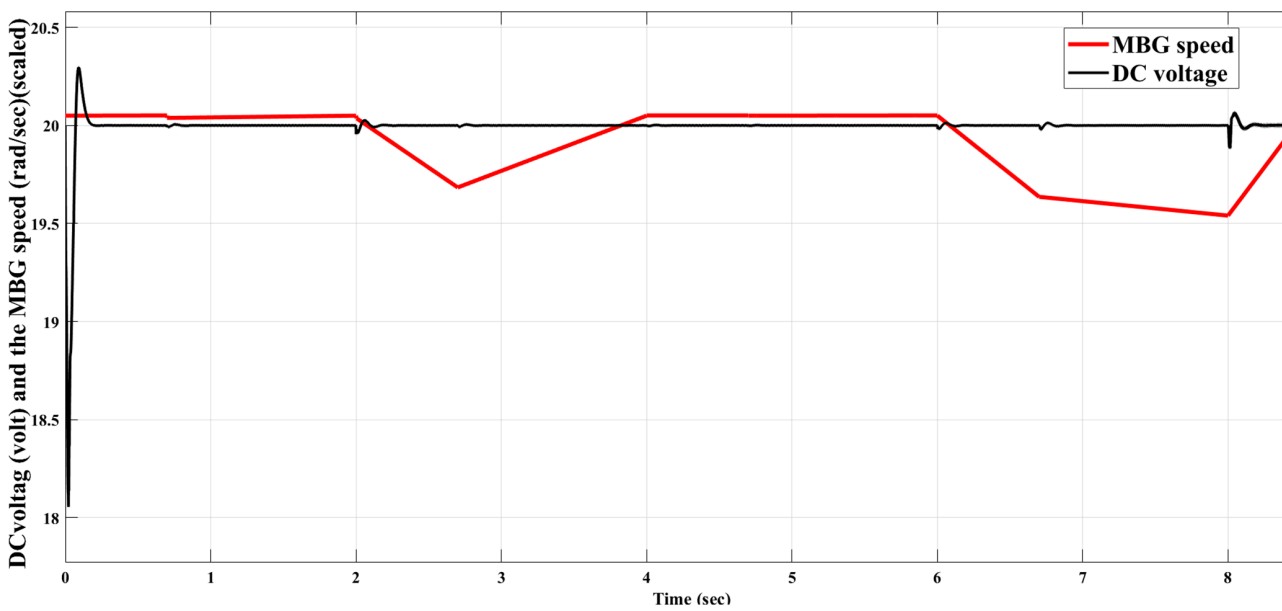

**Fig 5. Effect of MBG speed fluctuation disturbance on the output DC voltage using AFWATSMC.**

to the measured speed drive cycle of the grabber. Using the Signal Builder block in MATLAB Simulink, an emulation of this speed drive cycle was achieved. Although the speed fluctuates, the output DC voltage remains constant at 20 volts. This result demonstrates the robustness of the proposed AFWATSMC against an external disturbance.

Using the proposed AFWATSMC is essential for other types of disturbances. Fig 6 below demonstrates the performance of the proposed algorithm under the current and DC reference voltage variation for each battery. In Fig 6, the reference voltage of the controlled rectifier is varied according to the control signals generated by the designed Battery Management System (BMS). For example, if the State of Charge (SoC) of the 24-volt battery falls below the threshold defined by the BMS algorithm, the system adjusts the reference voltage of the AFWATSMC algorithm to 28 volts. Similarly, if the SoC of the 12-volt battery is low, the BMS sends a control signal to change the reference voltage to 15 volts. Simultaneously, the load condition of each battery was monitored to verify whether it meets the predefined loading criteria. This figure demonstrates the robustness of the proposed algorithm against various types of disturbances like reference voltage variation and load fluctuation.

## Results convergence check under parameters uncertainties and initial conditions variation

Table 3 is the result of the convergence check using the proposed AFWATSMC.

## Results for the AFWATSMC algorithm

Fig 7 below indicates the DC output voltage, error, and the reference direct axes current using free-will arbitrary convergence time ($T_f$) = 0.25. In this test, the convergence time was selected to be 0.2 s according to the user's will, and since it is clear that the DC voltage was convergence to the required reference value (20-volt), while the system state (DC voltage error) and the DC voltage will be close to zero starting from the selected convergence time.

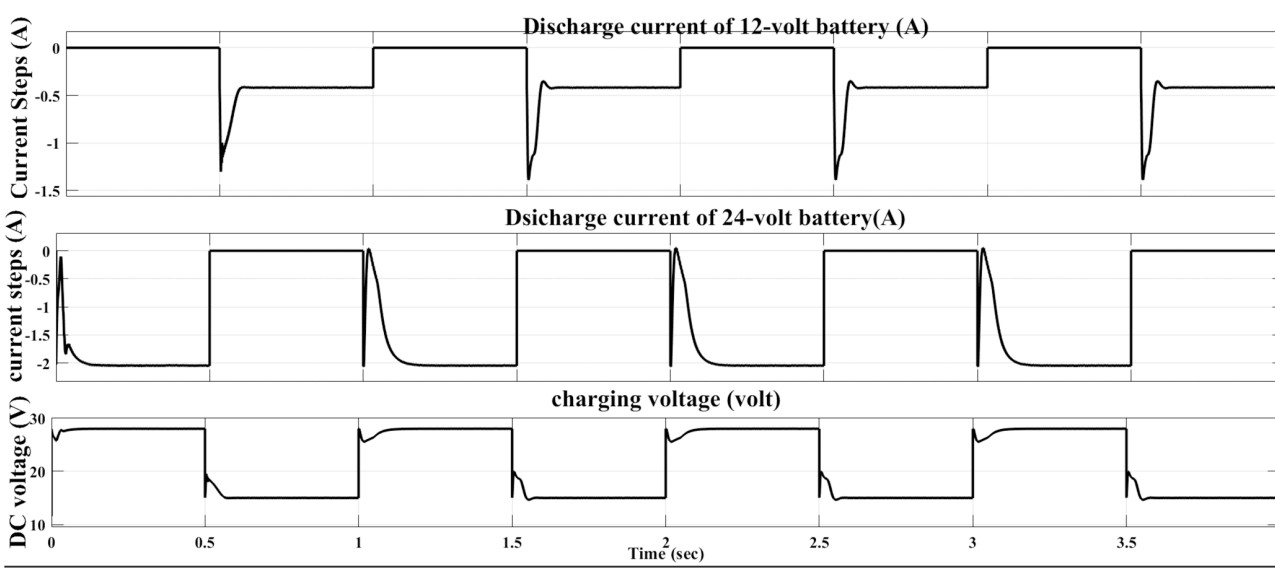

**Fig 6. Load and reference voltage fluctuation disturbance rejection feature of AFWATSMC.**

**Table 3. Convergence of the state under different parameter and initial condition variations using the FWAT algorithm.**

| Parameters/initial conditions | Parameters/initial condition variation % | | | |
|---|---|---|---|---|
| Inductance ($L$) | 30% | 30% | -30% | -30% |
| Capacitance ($C$) | 30% | 30% | -30% | -30% |
| Resistance ($R$) | 30% | 30% | -30% | -30% |
| Capacitor initial voltage ($v_C(0)$) | 100% | 120% | -100% | -120% |
| Inductor initial current ($i_L(0)$) | 100% | 120% | -100% | -120% |
| Does the state converge in FWAT? | yes | yes | yes | No |

## Parameter tuning results for the proposed AFWATSMC and DPSMC controllers

After the optimization of the AFWATSMC and DPSMC controllers using GA and PSO, the results of the control parameters were indicated in Table 4 below.

Table 5 below shows the multi-objectives PSO optimization results for parameter tuning of the current outer loop of the DPSMC algorithm.

## Validation results

This section demonstrates the result for the comparison between the proposed algorithm and another adaptive SMC algorithm.

**The AC output current of the alternator results.** Fig 8 below shows the phase (A) results of the AC output current of the alternator using the proposed algorithm and the ASMC. In this result, the AC waveform is more distorted in the ASMC case than the waveform generated using the new algorithm. This distortion affected the THD and harmonic behavior of the alternator. From the zoomed part of this figure , it is clear that the AC current signal obtained using the new algorithm is less distortion compared to the current signal obtained using the ASMC algorithm.

**THD value results.** Figs 9 and 10 below represents the FFT analysis of the AC of the alternator using the proposed AFWATSMC and the ASMC algorithms, respectively. The THD percentage in the controller using the new algorithm is 1.77%, but it is 3.34% in the case of the ASMC, indicating the robust response of the controller to achieve the least THD content.

**AC voltage results.** The AC voltage of the alternator was measured for both algorithms, as demonstrated in Fig 11 below. As shown in this figure, the new algorithm does not affect the AC voltage output of the alternator; the same response has been obtained for both algorithms. This figure demonstrates that both algorithms produce nearly identical AC voltage waveforms with no observable distortion or deviation, indicating that the proposed AFWATSMC algorithm does not interfere with the alternator's voltage generation characteristics.

**DC output voltage and control signal current of the voltage loop results.** Fig 12 below shows the rectifier output DC voltage using both algorithms. In the convergence time equal to 0.2 sec (as the user will), this DC voltage converges to its reference value within this time in the case of the AFWATSMC. In contrast, it asymptotically converges to the reference value in the case of the ASMC.The AFWATSMC curve shows minimal overshoot and negligible oscillations, reflecting improved transient performance.In contrast, the ASMC exhibits a slower settling time and a more noticeable ripple during the transient phase. Fig 13 below presents the behavior of the voltage loop's control signal $i_d^*$, that directly affects how the rectifier responds to voltage errors. The AFWATSMC control current reaches the target value more quickly (in 0.2 sec and exhibits a sharper rise, indicating a more decisive control effort

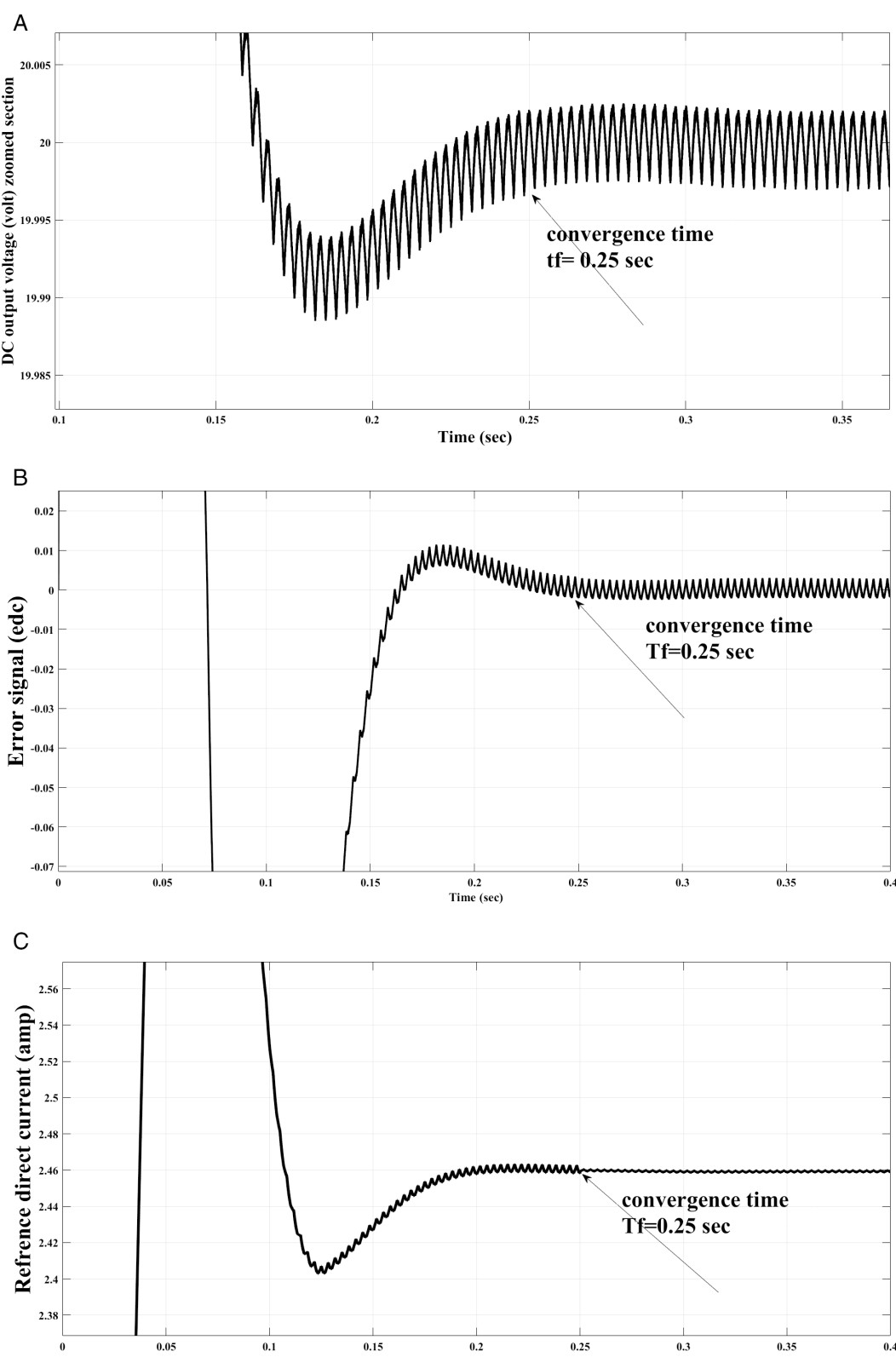

**Fig 7. DC output voltage (top left), Error of the DC output voltage (top right) and control signal (bottom) at convergence time of $T_f$ = 0.25 s.**

**Table 4. Results of Single Objective GA and PSO Parameters Tuning for the Voltage Loop of AFWATSMC.**

| DC voltage (V) | $k_1$ | $k_2$ | $\eta_1$ | $R(\Omega)$ | $L(H)$ | $C(F)$ |
|---|---|---|---|---|---|---|
| 15 | 233.075596 | 0.194039 | 0.076022 | 4.4325 | 5.05 mH | 5 mF |
| 20 | 233.075596 | 0.194039 | 0.076022 | 2.25 | 5.05 mH | 5 mF |
| 24 | 233.075596 | 0.194039 | 0.076022 | 1.25 | 5.05 mH | 5 mF |
| 28 | 233.075596 | 0.194039 | 0.076022 | 0.5 | 5.05 mH | 5 mF |

*Note.* The table presents the optimized parameters of the voltage loop controller using GA and PSO for different DC reference voltages.

**Table 5. Results of Multiobjective PSO Tuning for Current Loop DPSMC Controller.**

| Reference voltage (V) | G1_p | G2_p | G1_q | G2_q |
|---|---|---|---|---|
| 15 | 0.2129*1000 | 0.0372*1000 | 5.7694*1000 | 0.1023*1000 |
| 20 | 1.0403*1000 | 0.0036*1000 | 2.5373*1000 | 0.0280*1000 |
| 24 | 0.2129*1000 | 0.0372*1000 | 5.7694*1000 | 0.1023*1000 |
| 28 | 1.0e+03 * 5.7786 | 1.0e+03 * 0.0506 | 1.0e+03 * 5.7526 | 1.0e+03 * 0.0508 |

*Note.* This table summarizes the optimized DPSMC current loop controller parameters using multiobjective PSO under different reference voltage conditions.

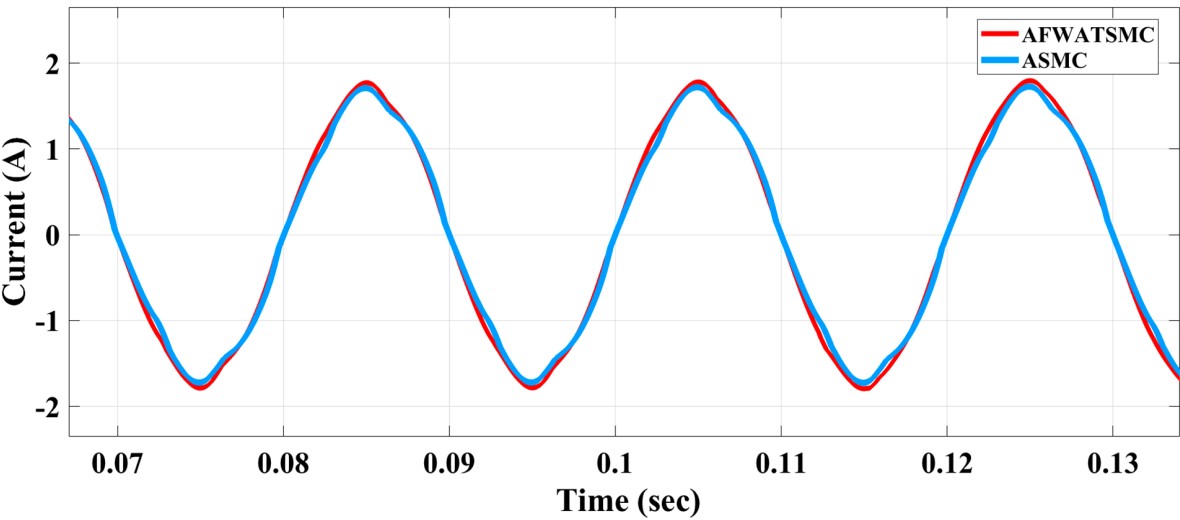

**Fig 8. AC output current waveform comparison between AFWATSMC and ASMC algorithms.**

in compensating for the voltage deviation.Meanwhile, ASMC responds more gradually (more than 0.5 sec), which explains the delayed stabilization of the DC voltage observed in Fig 13.

**Ripple factor results.** Fig 14 below demonstrates the difference in the ripple factor for both algorithms.This figure provides a comparative visualization of the DC output voltage ripple observed in the AFWATSMC and ASMC-controlled systems. As shown, the AFWATSMC controller significantly reduces voltage ripple once steady-state was achieved, as evidenced by the denser and more stable red waveform region. In contrast, the ASMC output (blue waveform) displays a broader oscillation envelope, indicating a higher ripple magnitude.

**Results on the convergence behavior of the voltage loop.** DC voltage error convergence behavior and the sliding surface behavior for the voltage loop were presented in Figs 15 and 16 below, respectively. In Fig 15, the AFWATSMC controller exhibits a faster reduction in

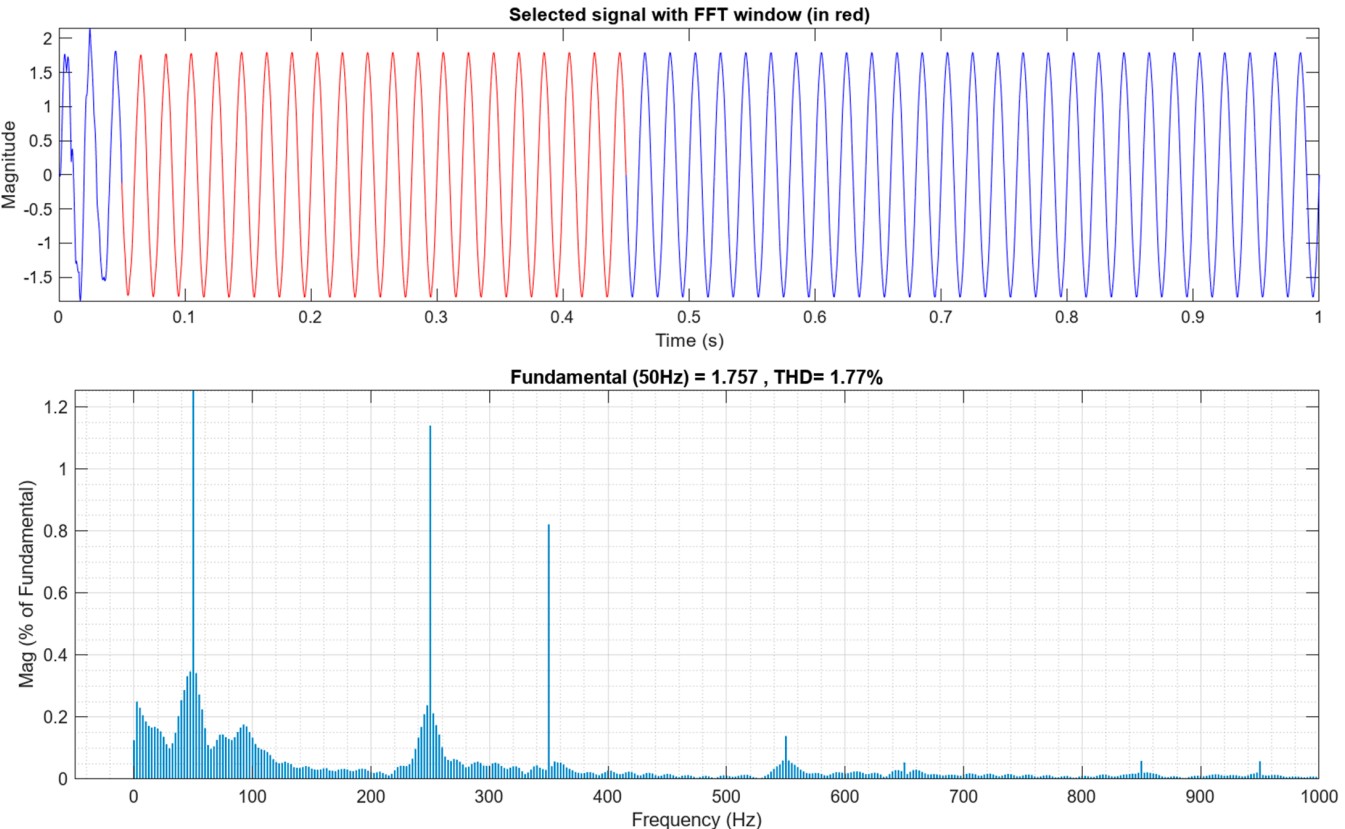

**Fig 9. FFT Analysis for the AC Output Current of the Alternator for AFWATSMC.**

error magnitude, reaching near-zero error within approximately 0.1 seconds. In contrast, the ASMC algorithm demonstrates a slower error decay, taking longer to stabilize. In Fig 16, The sliding surface under ASMC continuously increases, indicating an ongoing effort by the controller to force the system trajectory onto the sliding manifold. In contrast, AFWATSMC quickly stabilizes the sliding surface at a lower, steady value. This behavior highlights a key benefit of the proposed AFWATSMC approach: its adaptive and arbitrary-time convergence capability allows for earlier and smoother stabilization of system dynamics.

**Power factor results.** The power factor result was presented in Fig 17 below. The power factor measures how effectively the load utilizes the input power, with unity (1.0) indicating the ideal power conversion. The AFWATSMC algorithm exhibits faster recovery and steadier power factor convergence than ASMC. Despite initial transients, the AFWATSMC maintains the power factor very close to unity throughout the observation period, whereas ASMC shows more pronounced fluctuations and slower stabilization.

**Convergence behavior results of the current loop DPSMC controller.** Figs 18 and 19 present the active power behavior of the inner current loop controller using the two algorithms. In Fig 18, it is clear that the active power error using the new algorithm converges to zero within the convergence setting time, but it needs more time to be zero when using the ASMC. In Fig 19, the same behavior from the sliding surface is seen using the AFWATSMC; that is, the sliding surface converges to zero within the free-will time while it needs more time in the case of ASMC.

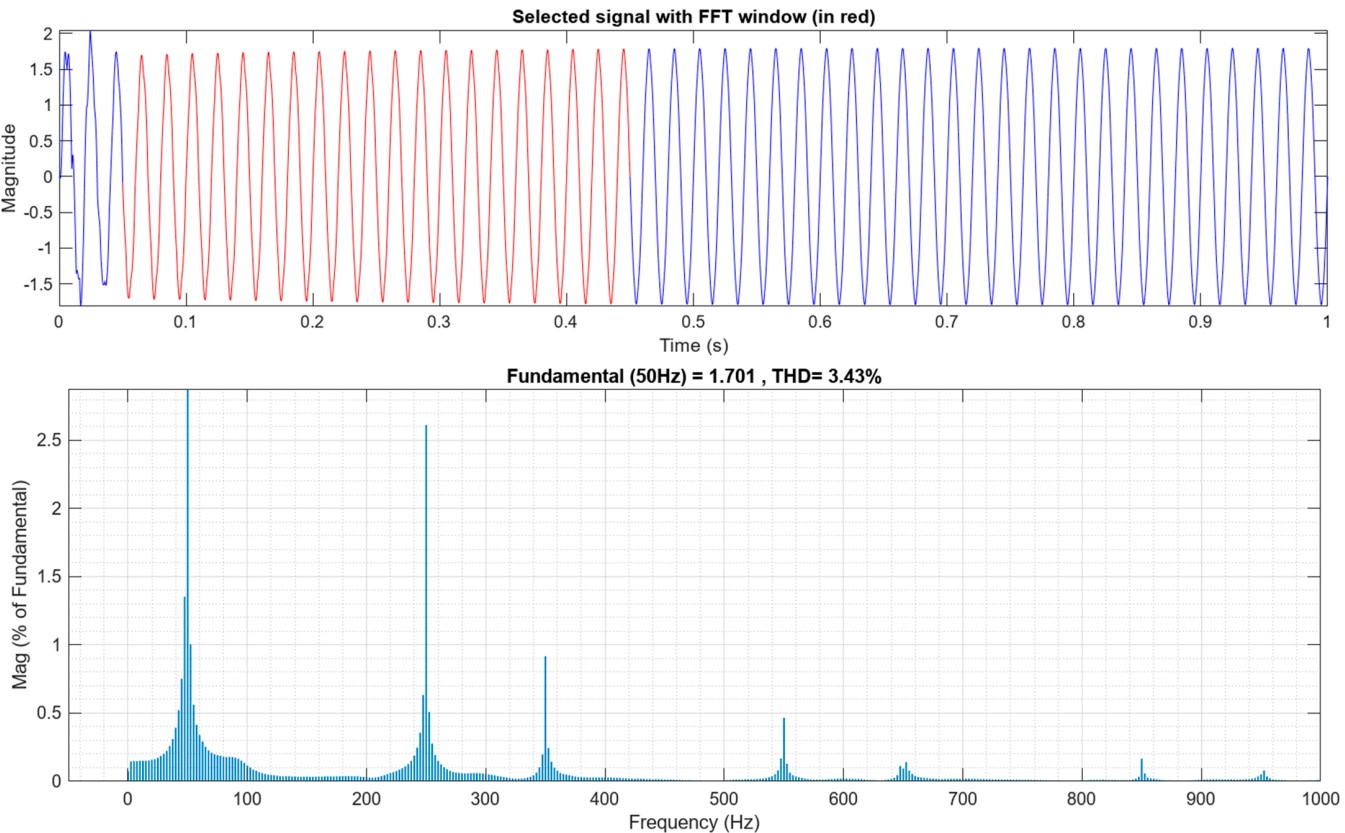

**Fig 10. FFT Analysis for the AC Output Current of the Alternator for ASMC.**

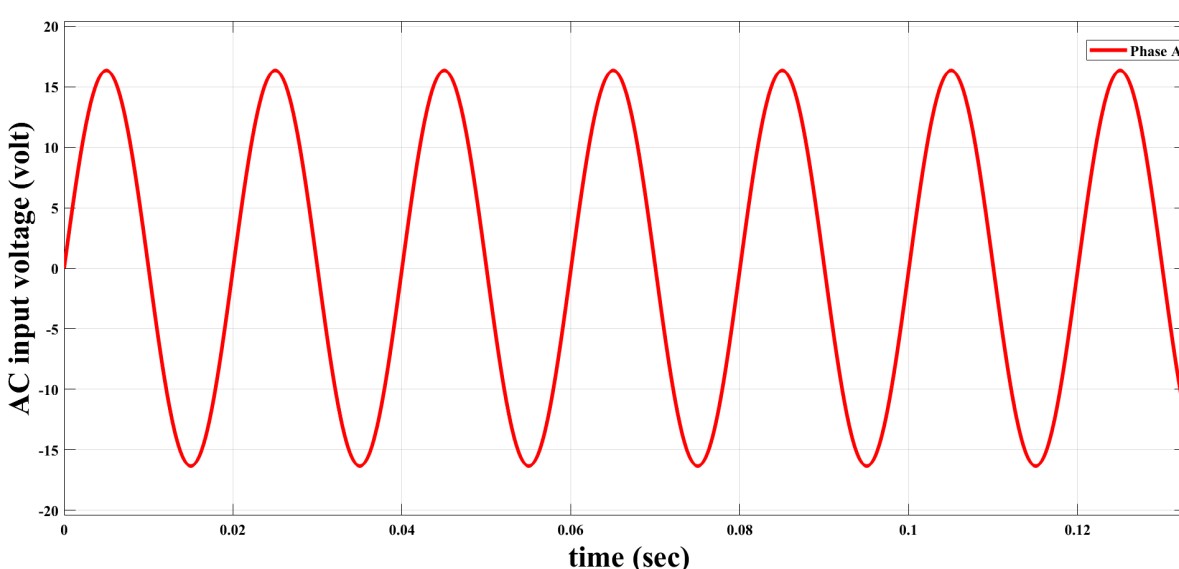

**Fig 11. Output AC voltage waveform of phase A under both AFWATSMC and ASMC control algorithms.**

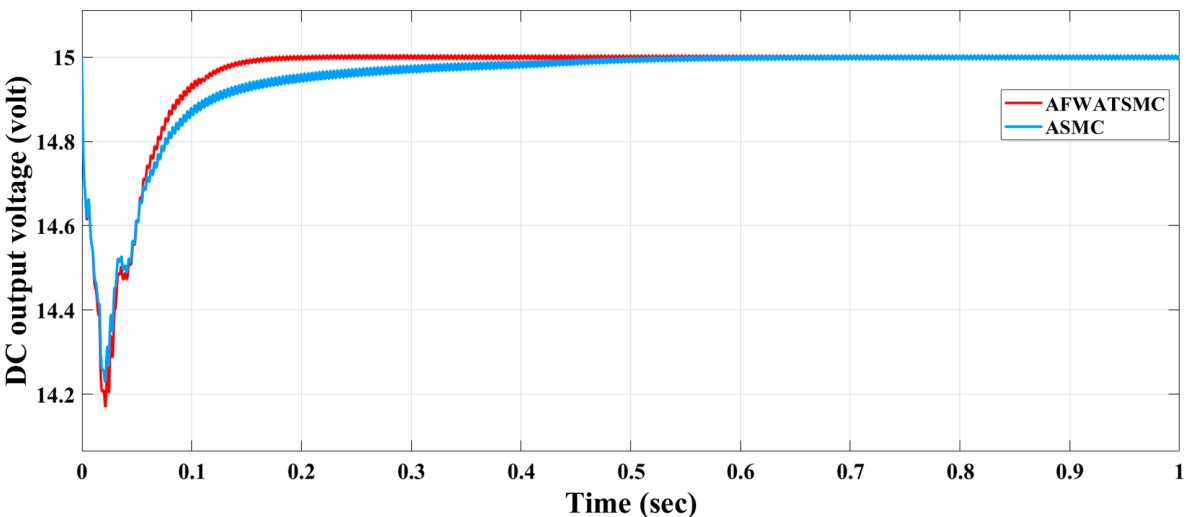

**Fig 12. Output DC voltage response comparison under AFWATSMC and ASMC controllers.**

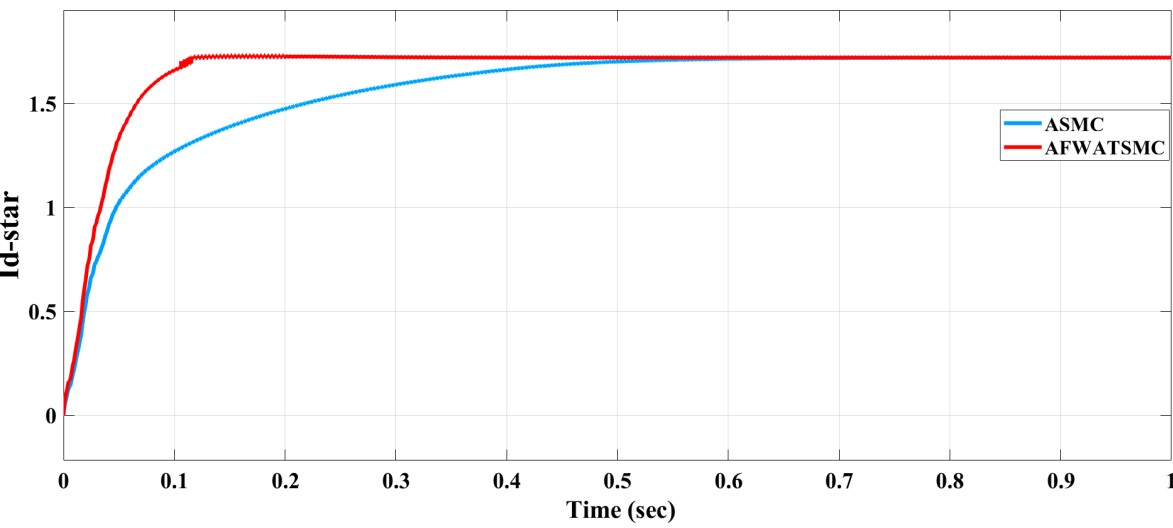

**Fig 13. Comparison of control signal response ($i_{dc*}$) for AFWATSMC and ASMC algorithms.**

**DC voltage behavior using both algorithms when reference voltage has different values.**
Fig 20 below compares the AFWATSMC and the ASMC's response when different loading and DC voltage references are used in the proposed algorithm with the BMS. Multiple tests—such as load variation and reference transitions between 28V, 15V, and back to 15V—are applied over a 4-second window to simulate realistic operational disturbances.

The AFWATSMC consistently demonstrates rapid settling times, lower overshoot, and better tracking precision than ASMC during each voltage transition. This can be clearly observed during sharp changes at t = 0.5 s, 1.2 s, 2.0 s, and 3.4 s. The zoomed-in inset provides a magnified view of the system behavior around one of the transition points (near t = 1.2 s), showing that AFWATSMC reduces oscillations and reaches the new setpoint more quickly and smoothly than ASMC.

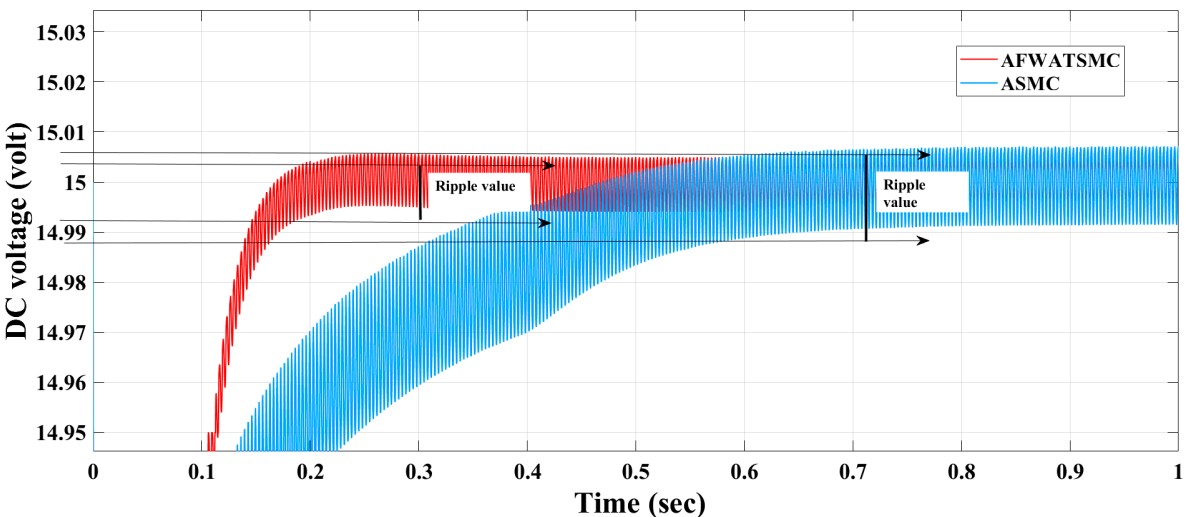

**Fig 14. Comparison of ripple values in zoomed-in DC output voltage for AFWATSMC and ASMC.**

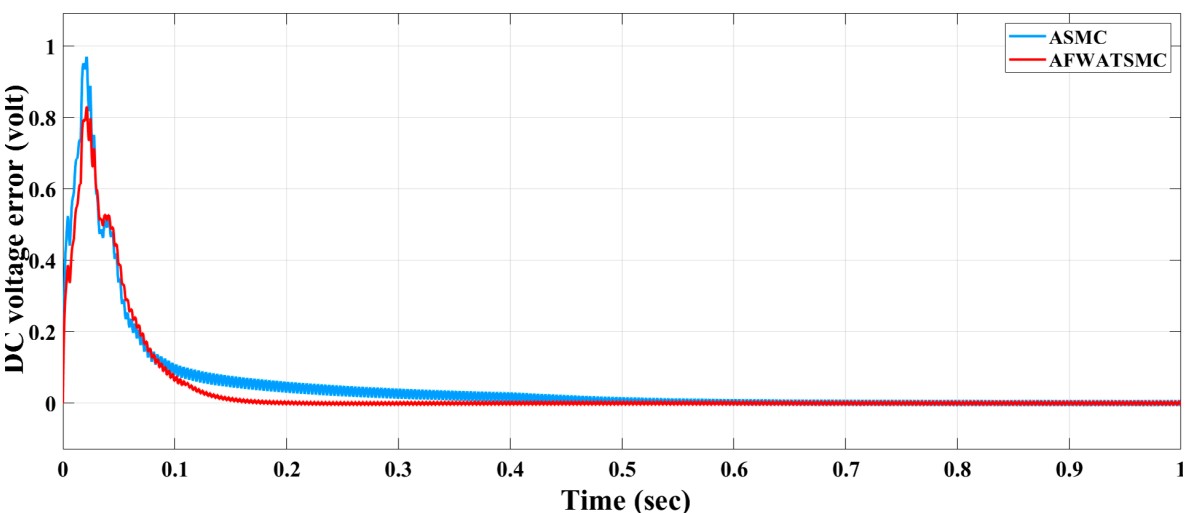

**Fig 15. Comparison of DC output voltage error signal for AFWATSMC and ASMC algorithms.**

**Alternator temperature behavior using both algorithms.** Using the thermal port on the Simscape alternator, the temperature behavior of the two algorithms has been tested as presented in Fig 21 below.

**Step response results for the DC output voltage for both algorithms.** Finally, a steady-state response was made for the DC output voltage for both algorithms, and the results were presented in Table 6 below.

## Comparative analysis with existing sliding mode control approaches

To emphasize the novelty and performance advantages, Table 7 below exhibits a comparative analysis of recent related works. The comparison is based on key technical parameters such as control strategy, response time, total harmonic distortion (THD), and robustness. This

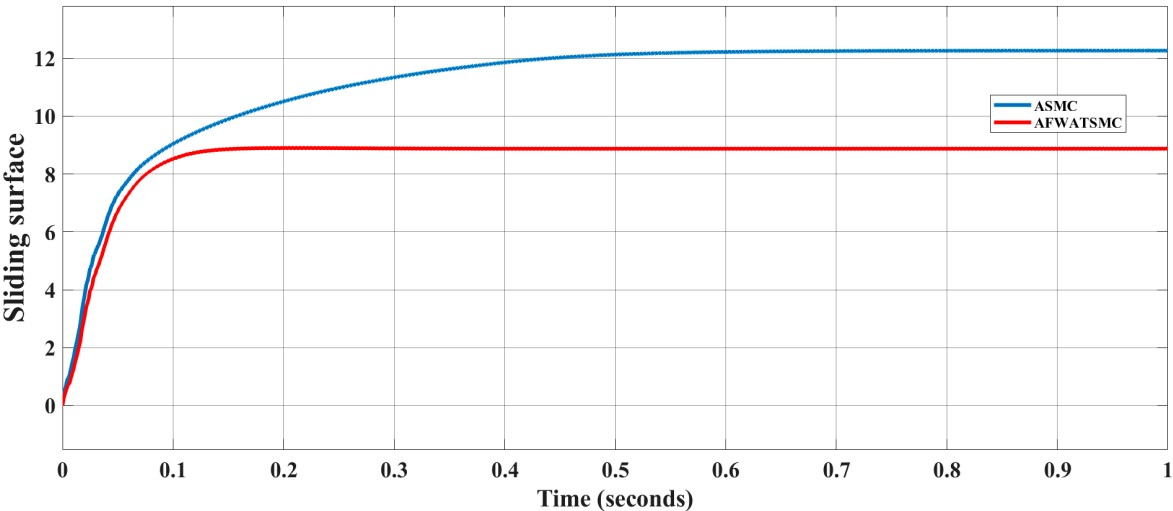

**Fig 16. Comparison of Sliding Surface Behavior for AFWATSMC and ASMC Algorithms.**

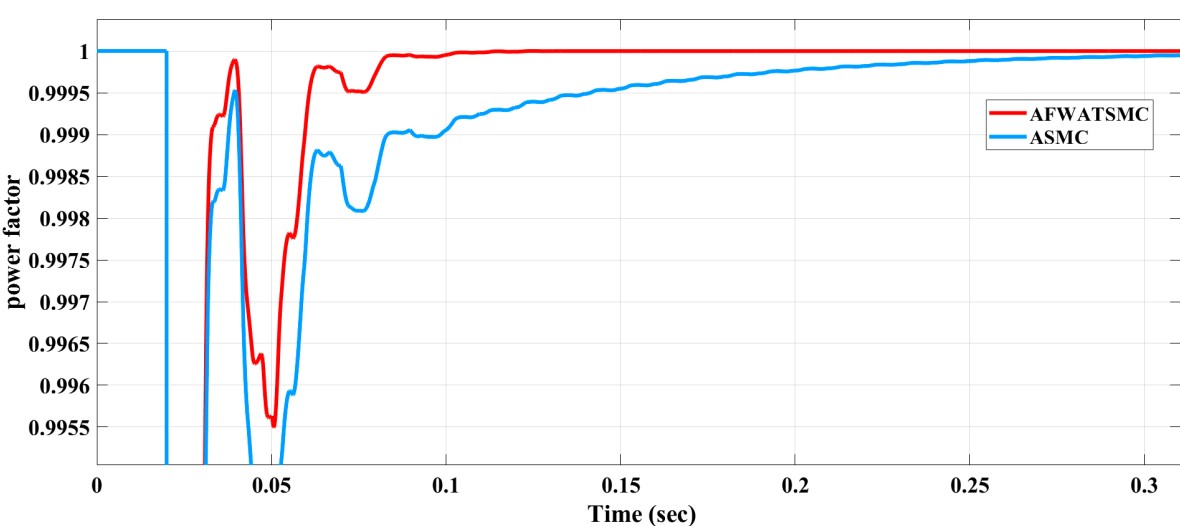

**Fig 17. Power Factor Performance Comparison Between AFWATSMC and ASMC Algorithms.**

table underscores the performance enhancements achieved in this study relative to existing approaches. This table is divided into two main sections; the first section discusses the hybrid approaches based on SMC applied to control the three-phase rectifier, and the second section demonstrates the pure SMC algorithms used with this type of rectifier. It is finally concluded that the FWAT approach has not yet been used to control the performance of a three-phase rectifier, to the best of current knowledge. The research gap has been identified through a recent review paper [52], which provides a detailed analysis of the methods and algorithms employed to control the performance of three-phase rectifiers based on SMC. The conclusion of this review clearly states that no previous work has implemented the Free-Will Arbitrary Tracking (FWAT) controller in power electronic applications—especially not for three-phase rectifiers.

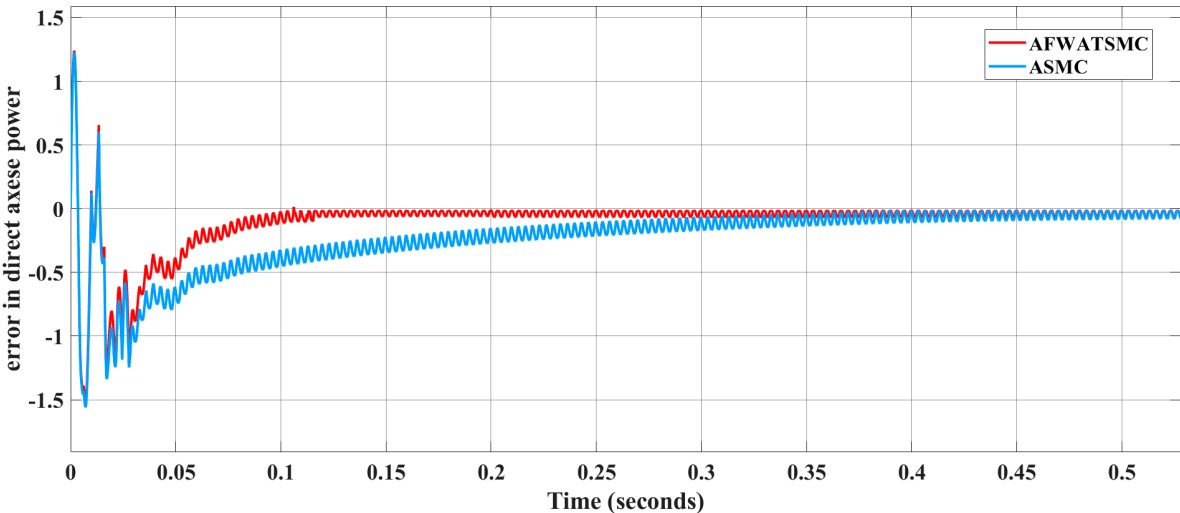

**Fig 18. Active Power Error Comparison Between AFWATSMC and ASMC Algorithms under DPSMC algorithm.**

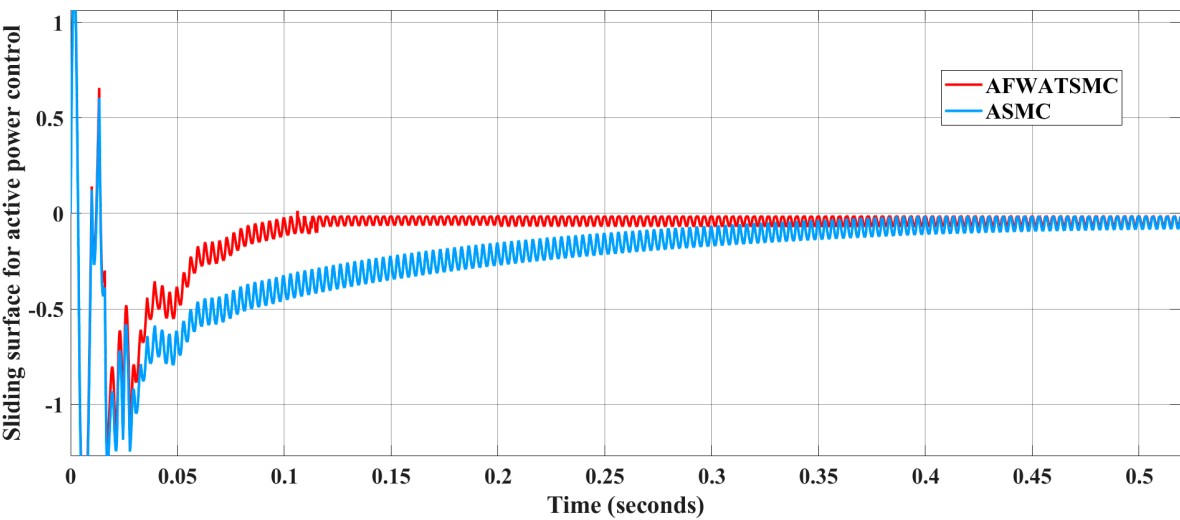

**Fig 19. Sliding Surface Response for Active Power Control Using AFWATSMC and ASMC.**

## Discussion

Using the proposed AFWATSMC can address this type of disturbance during the MBG's duty cycle and variable speed performance. The load variation and DC reference voltage fluctuation are other sources of disturbances, and this algorithm successfully addresses these disturbances. Table 3 is a sample of results when the parameters and the initial condition are varied simultaneously, and the system error converged to zero within $\pm 30\%$ fluctuations for all the parameters and $\pm 100\%$ for the initial conditions. As shown in Fig 7 above, the error converges to zero in 0.25 sec, and the DC voltage stops fluctuating. Fig 8 demonstrates the AC current behavior for the MBG alternator using the proposed algorithm and the ASMC, and it is clear that the current shape is more sinusoidal when using the new algorithm. In Fig 9, the integration of the algorithm and the filters reduces the THD value of the AC output current of

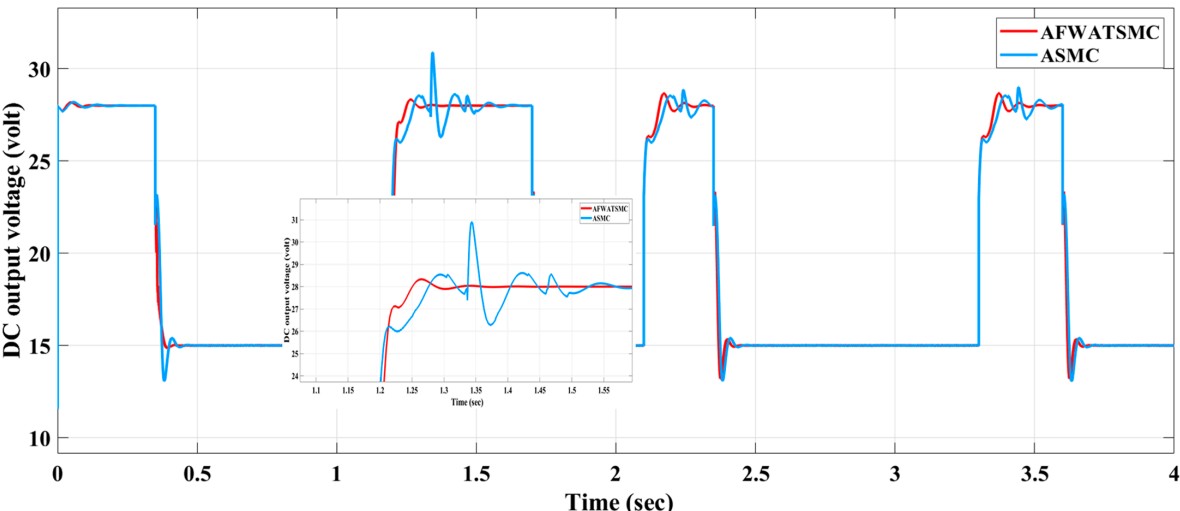

**Fig 20. Comparison between the response of the AFWATSMC And the ASMC when different loading and different DC volt reference are applied.**

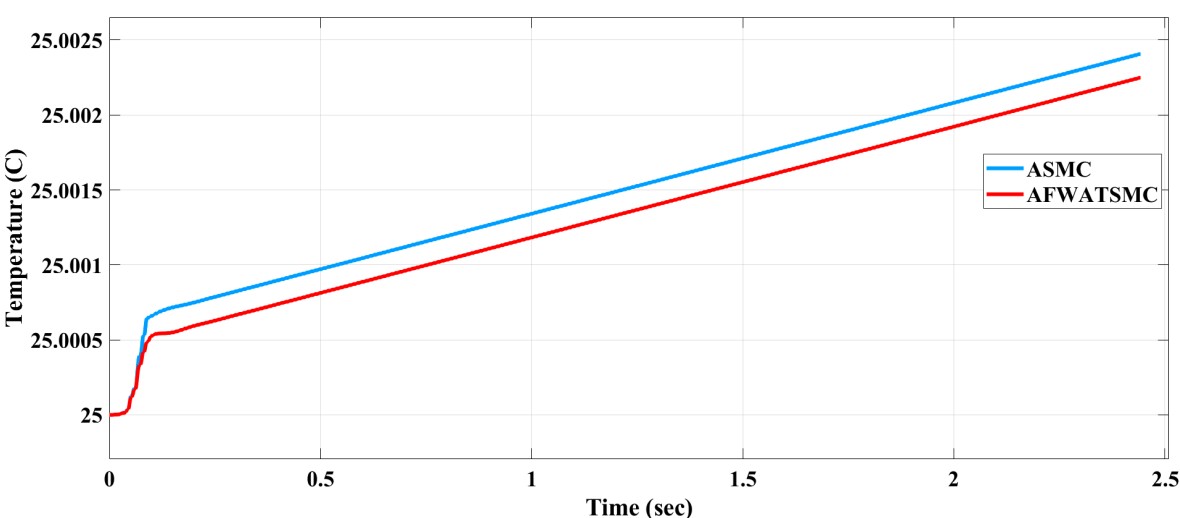

**Fig 21. Comparison between the thermal response of the alternator using AFWATSMC and ASMC.**

the alternator from 3.32% in the case of the ASMC to 1.77% in the case of the newly proposed algorithm. As presented in Fig 10, there was no enhancement or improvement in the AC voltage shape using the proposed algorithm. As it is seen in Fig 11, convergence to zero error was set at 0.2 seconds by the proposed algorithm (free-will arbitrary convergence), while it requires more than 0.6 seconds to converge to zero error with ASMC, the same was true for the control signal ($i_{dc}^*$). In the zoomed output DC voltage response in Fig 12, the ripple in the DC output voltage decreases from 2.5% using ASMC to 1.5% under the proposed new algorithm. From Fig 16, the unity power factor was obtained using the new algorithm, while the near unity power factor (0.998) was obtained using the ASMC algorithm. As shown in Fig 19, the new proposed algorithm is more robust against the reference variation disturbance than

**Table 6. Comparison of Dynamic Performance Parameters between AFWATSMC and ASMC Controllers.**

| Parameter name | AFWATSMC results | ASMC results |
|---|---|---|
| Rise Time | 0 | 0 |
| Transient Time | 0.1618 | 0.5284 |
| Settling Time | 0.0559 | 0.0613 |
| Settling in | 14.1689 | 14.2274 |
| Settling ax | 15.0058 | 15.0050 |
| Overshoot | 0.0068 | 0.0018 |
| Undershoot | 0 | 0 |
| Peak value | 15.0058 | 15.050 |
| Peak Time | 0.2665 | 0.9800 |

*Note.* This table compares the performance metrics of the AFWATSMC and traditional ASMC controllers in terms of response speed, overshoot, and peak characteristics.

the ASMC algorithm. From the temperature test results in Fig 20, it is clear that the temperature rise of the alternator connected to the rectifier that employs the AFWATSMC algorithm was less the one observed in the ASMC alternator because of the alternator's reduced THD

**Table 7. Comparative Summary of Three-Phase Rectifier Control Methods Based on Sliding Mode Control (SMC).**

| No. | Reference | Approach Name | Framework Description | Advantages | Limitations |
|---|---|---|---|---|---|
| 1 | [71,82] | Direct Power Control based SMC | Creation of sliding surface to track power goals. SMC forces system state onto this surface. | Acceptable performance, convergence to sliding surface, fixed frequency, less chattering (integral SMC). | complex, parameter tuning required ,and asymptotic Convergence time. |
| 2 | [83] | Backstepping based SMC | Backstepping creates global stability path, with SMC for disturbance rejection. | Simple implementation, experimental verification. | Complex tuning,chattering , not unity PF,and asymptotic Convergence time. |
| 3 | [81] | Virtual Synchronous Motor-based SMC | This approach uses a three-phase PWM rectifier modeled as a VSM combined with SMC. | Fast response, slight overshoot, and less steady-state error. | Simulation only with simplifications, chattering, distorted input current, complex, and asymptotic Convergence time. |
| 4 | [84] | Hybrid SMC with port-controlled Hamiltonian (PCH) | Reduces energy loss by guiding the system to a desired equilibrium with a PCH structure for energy storage and an SMC layer. | Robust against the system variations and good power factor. | Complex parameter estimation, shows unsatisfactory voltage/current responses, and lacks practical verification, and asymptotic Convergence time. |
| 5 | [85] | Hybrid State-Feedback and Sliding-mode Control | It employs state feedback to monitor voltage, current,and other system states. | Robust and faster dynamics than double PI control, achieving near-unity power factor and flexible active-to-reactive power adjustment. | complex, relies on accurate modeling and parameter tuning and approximate finite-time convergence. |
| 6 | [86] | Virtual Flux-Oriented Control based SMC (VFOC-ISMC) | It estimates stator flux from voltage and current data to guide rectifier control.SMC dynamically adjusts the PWM switching. | Avoids using PLL, shows strong performance, and maintains robustness against DC-bus load variations. | Sensitive to disturbances, complex,output ripple,simulation only, and asymptotic Convergence time. |
| 7 | [87,88] | Hybrid DPC hysteresis band with SMC. | Hysteresis band establishes allowable changes from a setpoint,SMC correct errors outside this band. | Experimentally validated, low THD, supports bidirectional power flow, unity power factor. | complex power calculations, multiple sensors, intricate switching logic, lacks finite-time convergence, and exhibits chattering. |
| 8 | [89] | Hybrid Linear Feedback Structure based SMC (LFS-SMC) | LFS ensures stable baseline control, while SMC enhances robustness by swiftly correcting deviations under disturbances and nonlinearities. | robust, adaptive, and flexible control with bidirectional power flow and has been validated experimentally. | Complex parameter tuning, no finite-time convergence,chattering in the output voltage. |

(*Continued*)

**Table 7.** (Continued)

| No. | Reference | Approach Name | Framework Description | Advantages | Limitations |
|-----|-----------|---------------|---------------------|------------|-------------|
| 9 | [90] | Hybrid Model Predictive Control based SMC (MPC-SMC) | MPC predicts future outputs and optimizes control actions, while SMC ensures robust and rapid error correction. | Offers robustness and simplicity, with adaptive parameter selection, reduced chattering, improved convergence speed. | Lacks finite-time convergence, involves a complex combination with many tunable parameters, and faces limitations in nonlinear systems. |
| 10 | [91] | Hybrid $H_\infty$ technique based STSMC ($H_\infty$-SMC) | $H_\infty$ stabilizes the voltage loop under disturbances, SMC ensures smooth and robust current loop. | Offers superior performance and robustness compared to PI, especially under varying load conditions. | The controller design is complex and requires parameter tuning for both inner and outer loops. No finite-time convergence nor experimental work. |
| 11 | [92] | Hybrid feedback/feedforward SMC | Feedback for real-time disturbance correction and feedforward for proactive adjustment. | links output power and negative incremental impedance, stabile by Lyapunov. | Simulation only,no analysis of load variations,chattering,no comparison with other strategies, and Finite-Time Convergence. |
| 12 | [93] | Observer-based SMC | SMO used to estimate critical system states that are not directly measurable, like voltage. | Provides robust performance, fast convergence, fewer sensors, and reduced complexity. | disturbance-sensitive, parameter tuning requirement, depends heavily on model accuracy, and Finite-Time Convergence. |
| 13 | [94] | Hybrid Fuzzy based SMC (FSMC) | Fuzzy logic modifies SMC parameters in real-time to optimize the control laws and sliding surface. | Reduced chattering via FSMC, robust, fast response, and validated practically. | complex parameter tuning, requires computation, sensitive to model uncertainties, and asymptotic convergence. |
| 14 | This work | AFWATSMC | used a novel AFWATSMC and the second loop used DPSMC | Design a novel approach to guarantee the error convergence within time set by the user will regardless of the system disturbances, and nonlinearities | Lake of experimental validation and generality of applications. |

*Note.* The table summarizes different SMC-based methods applied to three-phase rectifier control systems, highlighting each method's framework, strengths, and drawbacks.

value and reduced loading effect. Finally, the transient time of the proposed algorithm was significantly shorter than that of the ASMC algorithm. The settling time was slightly less in the case of the proposed algorithm. The AFWATSMC reaches its final stable state faster than the ASMC. The peak time value was reached faster with the AFWATSMC algorithm within ASMC. However, both algorithms quickly responded to the rise time measurement and zero undershoot. Regarding overshoot, the ASMC was marginally lower than the proposed algorithm. AFWATSMC reaches its peak value considerably quicker than ASMC showcasing its rapid control action.

## Conclusion

In this study, a high level of precision and robustness was achieved using the AFWATSMC with the DPSMC algorithms to control the performance of the controlled rectifier in place of the uncontrolled rectifier in the MBG DC voltage system. Applying this algorithm to the MBG DC voltage system enabled efficient handling of significant disturbances and uncertainties. The proposed control rectifier accommodates the fluctuations in the AC voltage of the alternator arising from the fluctuations in the MBG speed. Furthermore, the controller compensates the disturbance due to the variation of the various MBG loads and the changes in the

reference voltage that have been changed according to the DC voltage system requirements. These results highlight its reliability and adaptability under diverse operational conditions relative to existing literature. Overall, the proposed algorithm guarantees convergence within a free-will arbitrary time regardless of the initial conditions and system parameter variations. The proposed AFWATSMC algorithm successfully maintained convergence within free-will arbitrary time (($T_f$) =0.25 seconds) or any free-will time of convergence, with nearly zero error and no fluctuations in reference output DC voltage. This feature exhibits the system's capability of adapting to free-will convergence times, ensuring an accurate and stable performance that rejects various uncertainties and disturbances. Comparison of the AFWATSMC and the ASMC for the rectifier voltage loop indicates a notable improvement in the new algorithm's rectifier performance. The AC signal of the alternator current becomes more sinusoidal compared with the proposed algorithm. Furthermore, the THD value of the current was reduced to more than half of its value using the new algorithm with (46.96%) enhancement. The THD value of the proposed algorithm was below the commonly accepted thresholds outlined in IEEE Standard. Additionally, error convergence to zero occurs in the free-will arbitrary time with the proposed algorithm. By contrast, the convergence to zero in the ASMC algorithm requires more time, indicating an asymptotically stable controller. Moreover, an improvement in the ripple factor and power factor was observed with the proposed algorithm; the percentage improvement for the ripple factor was 66.6% using the zoomed fig of the DC voltage, and the improvement in the power factor was 0.2%. This behavior reduces the risk of MBG alternator stress and increases its lifespan. Moreover, the faster convergence to zero error, as demonstrated by the reduction in time from 0.6 seconds with the previous system (ASMC) to 0.2 seconds with AFWATSMC, reflects the system's enhanced capability to return to a steady state quickly after a disturbance. The reduction in alternator THD and loading effect lowers the alternator's temperature; this further confirms the proposed algorithm's effectiveness. From the step response of the DC voltage of the rectifier , the results in table 4 shows that the new algorithm outperforms ASMC by submitting faster transient and settling times, reduced overshoot percentage, and rapid peak response. Both controllers achieve same steady-state accuracy, yet AFWATSMC provides enhanced dynamic performance and robustness for DC voltage regulation in three-phase rectifiers. All in all , the core innovation of this study lies in the integration of a newly free-will arbitrary time of convergence algorithm with an adaptive sliding mode controller construction, allowing user-defined settling time regardless of system uncertainties, disturbances, or initial conditions—marking an important advancement over existing rectifier control approaches.

## Limitations of the study

This work seeks to design a robust and flexible controller to enhance the performance of a three-phase rectifier applied with a customized battery management system to manage the multi-battery DC voltage systems of the MBG. However, several limitations remain:

1. **Simulation-Based Validation Only:** The proposed AFWATSMC was validated exclusively through MATLAB programming and Simulink models. While MATLAB tools offer a highly controlled and repeatable analysis environment, they do not fully emulate the nonlinearities in the actual application, time delay issues, and measurement noise that can be found in real-world systems.

2. **Absence of Experimental Application:** Experimental validation was not performed because setting up such a system is time-intensive and resource-demanding. In addition, the complicated design of the filtering system for this specific hardware setup poses implementation challenges.

3. **Real-Time Feasibility and Embedded Integration:** The current control strategy has not yet been examined on a real-time platform. Although it demonstrates stability and robustness in MATLAB simulation, practical implementation may encounter computational complexity and real-time processing limitations, especially on embedded systems with constrained processing power and memory. Future work should include deployment on microcontrollers or FPGA-based platforms to assess real-time feasibility and integration potential.

4. **Computational Cost Issue:** The developed AFWATSMC algorithm includes adaptive mechanisms and optimization-based parameter tuning (using GA and PSO), which may lead to higher computational demand compared to conventional controllers. This could pose difficulties for systems requiring fast response times or operating on low-power embedded systems.

5. **Application Specificity:** This study focuses on the power system of an existing oil palm grabber, which may limit the generalizability of the proposed approach to other systems without significant adjustments.

6. **Limited Load Consideration:** Load consideration for the proposed MBG voltage system is another limitation. The AFWATSMC was verified using the specific electrical load profile of the MBG studied in this research, where the overall load was treated as a combination of autonomous subsystems, including actuators, sensors, controllers, and other power-consuming components.

## Acknowledgments

Sincere gratitude is extended to the Faculty of Engineering at Universiti Putra Malaysia (UPM) for their continuous support throughout this research. Special appreciation is given to the Power Electronics Laboratory and the Robotics Workshop within the faculty for providing invaluable resources, facilities, and technical assistance. Their support significantly contributed to the successful completion of this study.

## Author contributions

**Conceptualization:** Omar Talal Mahmood, Wan Zuha Wan Hasan.

**Data curation:** Omar Talal Mahmood, Wan Zuha Wan Hasan, Norhafiz Azis.

**Formal analysis:** Wan Zuha Wan Hasan, Nor Mohd Haziq Norsahperi, Norhafiz Azis.

**Investigation:** Omar Talal Mahmood, Wan Zuha Wan Hasan.

**Methodology:** Omar Talal Mahmood, Wan Zuha Wan Hasan, Hafiz Rahidi Ramli.

**Project administration:** Wan Zuha Wan Hasan.

**Resources:** Luthffi Idzhar Ismail, Nor Mohd Haziq Norsahperi, Norhafiz Azis.

**Software:** Omar Talal Mahmood, Wan Zuha Wan Hasan, Hafiz Rahidi Ramli.

**Validation:** Wan Zuha Wan Hasan, Luthffi Idzhar Ismail, Nor Mohd Haziq Norsahperi, Norhafiz Azis.

**Writing – original draft:** Omar Talal Mahmood, Wan Zuha Wan Hasan, Hafiz Rahidi Ramli.

**Writing – review & editing:** Wan Zuha Wan Hasan, Luthffi Idzhar Ismail, Hafiz Rahidi Ramli.

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
