## [Decision Letter · Decision Letter 0]

27 May 2025

PONE-D-25-20009Adaptive Free-Will Arbitrary Time Sliding Mode Controller for Three-Phase Rectifier Used in Oil Palm Autonomous GrabberPLOS ONE

Dear Dr. Mahmood,

Thank you for submitting your manuscript to PLOS ONE. After careful consideration, we feel that it has merit but does not fully meet PLOS ONE’s publication criteria as it currently stands. Therefore, we invite you to submit a revised version of the manuscript that addresses the points raised during the review process.

It is suggested to please address all concerns and suggestions of the reviewers point-wise and also improve or expand the article where needed in order to justify the responses to the concerns. Therefore a major revision is proposed. 

We look forward to receiving your revised manuscript.

Kind regards,

Akhtar Rasool, Ph.D.

Academic Editor

PLOS ONE

Journal Requirements:

4. Please amend the manuscript submission data (via Edit Submission) to include author Wan Zuha Wan Hasan.

5. Please amend your authorship list in your manuscript file to include author Wan Zuha Wan Hasan Wan Hasan.

6. Please remove your figures from within your manuscript file, leaving only the individual TIFF/EPS image files, uploaded separately. These will be automatically included in the reviewers’ PDF**.**

Additional Editor Comments :

It is suggested to please address all concerns and suggestions of the reviewers point-wise and also improve or expand the article where needed in order to justify the responses to the concerns. Therefore a major revision is proposed.

Reviewers' comments:

Reviewer's Responses to Questions

**Comments to the Author**

1. Is the manuscript technically sound, and do the data support the conclusions?

Reviewer #1: Yes

Reviewer #2: Partly

2. Has the statistical analysis been performed appropriately and rigorously? 

Reviewer #1: Yes

Reviewer #2: N/A

3. Have the authors made all data underlying the findings in their manuscript fully available?

Reviewer #1: Yes

Reviewer #2: No

4. Is the manuscript presented in an intelligible fashion and written in standard English?

Reviewer #1: Yes

Reviewer #2: No

5. Review Comments to the Author

Reviewer #1: The authors have proposed “Adaptive free-will arbitrary time sliding mode controller for three-phase rectifier used in oil palm autonomous grabber”. In this the reviewer is pleased to inform that the paper is well written and documented. However, some corrections are required for the improvement of the quality of the paper. It is suggested to follow the suggestions as mentioned below and submit the revised version for further consideration. Overall decision by the reviewer is MAJOR REVISION.

Suggestions are as follows:

(1) The Abstract is quite elaborative. Abstract must be the summary of the entire paper in a concise and compact form. Thus, it is suggested to make the abstract compact by mainly focusing on the novelty of the proposed work.

(2) The Introduction is good and the literature reviews are well highlighted. However, the entire Introduction is written in a single paragraph. It is suggested to break it in small paragraphs. In the last paragraph, the contribution of the proposed work along with the organization of the rest parts the paper is to be included.

(3) In Figure 1, the images just above the blocks “Three Phase Rectifier” and “BMS” are absolutely blurred and nothing is visible. It is suggested to include HD images for those two pictures.

(4) In the sentence on page no. 3: “The phase rectifier can be represented in the below three equations [28]:”, there are four equations (1-4) not three. Correct.

(5) Below equation (13), replace r and e by r and e.

(6) Below equation (13), there is a long text. Break in paragraphs.

(7) Below equation 31, the sentence: “The advantages of using such an addition are It will provide a free-will convergence to the error of the DC-link voltage of the rectifier, which is the desired state (Vdc →V∗dc) within the arbitrary time (Tfadjusted) regardless of initial conditions or disturbances.” is ambiguous and not clear. Rewrite it. It is suggested to write all advantages either by using bullets or by numbering.

(8) Place equation 8 at the center of the page.

(9) The font size of Eqn. 10 is not same like other equations. Correct Eqn. 10. Also place the equation number 10 at right aligned position.

(10) Below Fig. 6, the authors have written “Figure 6 illustrates the iq – iq harmonic current detection algorithm based on instantaneous reactive power theory.” However, the figure caption describes the block diagram. It is ambiguous.

(11) Fig 3 is blurred. Replace it by HD image.

(12) In step c:Error Threshold check of the Algorithm for the AFWATSMC for the sentence: “Next, the algorithm evaluate whether the error (edc) exceeds the predefined threshold set bt the algorithm” what “bt” means.

(13) On page 15 for the sentence: “Finally, the same 15 v as a reference voltage and loading conditions were applied to both systems.” 15 v will be replaced by 15 V.

(14) Explain Fig. 5 and Fig. 6 in more detail.

(15) Replace all figures of Fig. 7 with HD images and large font for axes indentations.

(16) Write all parameters in Table 2 in italics font.

(17) Change yellow color of all plots by some other deep colors like red or black. Also increase the depth of colors to understand the comparison plots between AFWATSMC and ASMC.

(18) Fig. 9 is absolutely blurred. It can’t be accepted for the paper. All left hand side texts are blurred. Replace Fig. 9 by HD images. It is suggested to separate the images and place individually one after another with proper figure number.

(19) Write concluding remarks based on Table 4.

(20) Include a comparative table with some recently published works based on some common parameters to justify the novelty of the proposed work.

Reviewer #2: This manuscript proposes a novel Adaptive Free-Will Arbitrary Time Sliding Mode Controller (AFWATSMC) for a three-phase rectifier used in an oil palm autonomous grabber. The aim is to improve DC voltage regulation and reduce total harmonic distortion (THD) while addressing thermal issues due to excessive loading on the alternator in a robotic mobile platform. The paper includes an original control scheme with adaptive terms and modified convergence logic using MATLAB/Simulink modeling, genetic algorithms (GA), and particle swarm optimization (PSO) for parameter tuning. While the topic is timely and interesting, and the integration of FWAT with SMC is original, the manuscript requires significant revisions to improve readability, clarity, and alignment with standard engineering practices.

Major Comments

1. Structure and Flow:

• The manuscript is dense and lengthy, with long paragraphs and limited use of subheadings, making it difficult to navigate. Sections such as the introduction and methods should be broken down into clearly numbered subsections to improve organization and flow.

• The title is confusing and overly complex. Consider rephrasing to emphasize the application and novelty (e.g., "An Adaptive Sliding Mode Controller with Arbitrary-Time Convergence for Three-Phase Rectifiers in Autonomous Agricultural Vehicles").

2. Introduction:

• The introduction sets a solid context about the challenges in oil palm harvesting and the use of MBGs, but it lacks citations of recent work in control systems for autonomous agricultural machinery or rectifier control.

• While the motivation is clear, the novelty of the proposed controller should be stated explicitly. What differentiates this controller from other SMC variants or existing rectifier control strategies?

3. Novelty and Contribution:

• The key innovation lies in integrating free-will arbitrary convergence time with an adaptive SMC structure. However, this should be highlighted more clearly in the abstract and conclusion.

• The mathematical derivations are comprehensive, but the novelty gets buried in formula-heavy sections. A concise summary (e.g., bullet points or a table) comparing your approach to ASMC or other SMC methods would enhance clarity.

4. Figures and Tables:

• Many figures are not discussed adequately in the text. For example, Figures 5–16 are introduced with minimal analysis or interpretation.

• Captions are often too brief or generic. For example, “DC Output Voltage Behavior” doesn’t explain the scenario or what distinguishes the behavior shown.

• Table 1 on convergence lacks context—what does “yes” or “no” convergence mean practically?

• Consider adding a summary table of all tested conditions and system responses for clarity.

5. English Language and Style:

• The manuscript suffers from grammatical inconsistencies, awkward phrasing, and run-on sentences (e.g., the sentence starting with "The advantages of using such an addition are...").

• Please consider professional English language editing. Use consistent past tense for experimental descriptions and present tense for general truths.

• Terms like "free-will arbitrary time" are unconventional in engineering literature. You may want to explain this term clearly or use more standard terminology such as “user-defined convergence time.”

6. Validation and Discussion:

• While the controller is validated in simulation and parameter tuning is supported by GA and PSO, the physical implementation is not discussed. Is this purely simulated? If so, limitations should be explicitly stated.

• The paper would benefit from a quantitative comparison of key performance metrics (e.g., THD, convergence time, ripple factor) between AFWATSMC and conventional controllers.

• More insights should be provided into practical implementation issues: real-time feasibility, computational cost, and integration with embedded systems.

7. References:

• The literature review is broad but misses key studies on modern SMC applications in rectifiers, such as those using model predictive control, hybrid controllers, or grid-connected converters.

• The reference formatting is inconsistent; ensure all references follow PLOS ONE guidelines.

8. Suggested References (for Authors to Consider Including):

• Here are five relevant references that the authors might cite to strengthen the novelty and relevance of the harmonic mitigation and control methodology (especially for high-performance rectifier systems). These references would enhance the relevance of your harmonic mitigation discussion and modernize your bibliography:

1) Al-Barashi, M., Zou, A., Wang, Y., Luo, W., Shao, N., Tang, Z., & Lu, B. (2025). Magnetic Integrated Multi-Trap Filters Using Mutual Inductance to Mitigate Current Harmonics in Grid-Connected Power Electronics Converters. Energies, 18(2), 423. https://doi.org/10.3390/en18020423.

2) Al-Barashi M, Wang Y, Bhutta MS (2024) High-frequency harmonics suppression in high-speed railway through magnetic integrated LLCL filter. PLOS ONE 19(6): e0304464. https://doi.org/10.1371/journal.pone.0304464.

3) Al-Barashi, M., Wang, Y., Lan, B. et al. Magnetic integrated double-trap filter utilizing the mutual inductance for reducing current harmonics in high-speed railway traction inverters. Sci Rep 14, 10058 (2024). https://doi.org/10.1038/s41598-024-60877-y.

4) Al-Barashi, M., Meng, X., Liu, Z., Saeed, M.S.R., Tasiu, I.A., Wu, S.: Enhancing power quality of high-speed railway traction converters by fully integrated T-LCL filter. IET Power Electron. 16, 699–714 (2023). https://doi.org/10.1049/pel2.12415.

5) Al-Barashi, M., Wu, S., Liu, Z., Meng, X., Tasiu, I.A.: Magnetic integrated LLCL filter with resonant frequency above Nyquist frequency. IET Power Electron. 15, 1409–1428 (2022). https://doi.org/10.1049/pel2.12313.

Summary and Recommendation

• This manuscript presents a potentially impactful control strategy for improving power quality in robotic agricultural platforms. However, in its current form, the manuscript suffers from lack of clarity, verbosity, and insufficient comparative discussion.

Recommendation: Major Revision

• The paper has potential, but it requires significant reorganization, improved English usage, a stronger literature review, and clearer highlighting of contributions before being suitable for publication.

6. PLOS authors have the option to publish the peer review history of their article (what does this mean?). If published, this will include your full peer review and any attached files.

Reviewer #1: **Yes: **DR. TARUN KUMAR DAS

Reviewer #2: No

---

## [Author Response · Author response to Decision Letter 1]

13 Jun 2025

we did all the required correction from the editor and the reviewers

---

## [Decision Letter · Decision Letter 1]

13 Jul 2025

PONE-D-25-20009R1An Adaptive Sliding Mode Controller with Free-Will Arbitrary Time Convergence for Three-Phase Rectifiers in Autonomous Agricultural VehiclesPLOS ONE

Dear Dr. Mahmood,

Thank you for submitting your manuscript to PLOS ONE. After careful consideration, we feel that it has merit but does not fully meet PLOS ONE’s publication criteria as it currently stands. Therefore, we invite you to submit a revised version of the manuscript that addresses the points raised during the review process.

We look forward to receiving your revised manuscript.

Kind regards,

Akhtar Rasool, Ph.D.

Academic Editor

PLOS ONE

Journal Requirements:

**Additional Editor Comments:**

I congratulate the authors that the reviewers have acknowledged the improvements in the article and they are also satisfied with the responses submitted. However, they have raised a couple of points which we feel will be very important to raise the quality of your manuscript and as well as its presentation. So, we are advising a minor revision and expect to receive the final corrected version at the earliest. Thanks

Reviewers' comments:

Reviewer's Responses to Questions

**Comments to the Author**

1. If the authors have adequately addressed your comments raised in a previous round of review and you feel that this manuscript is now acceptable for publication, you may indicate that here to bypass the “Comments to the Author” section, enter your conflict of interest statement in the “Confidential to Editor” section, and submit your "Accept" recommendation.

Reviewer #1: All comments have been addressed

Reviewer #2: All comments have been addressed

2. Is the manuscript technically sound, and do the data support the conclusions?

Reviewer #1: Yes

Reviewer #2: Yes

3. Has the statistical analysis been performed appropriately and rigorously? 

Reviewer #1: Yes

Reviewer #2: N/A

4. Have the authors made all data underlying the findings in their manuscript fully available?

Reviewer #1: Yes

Reviewer #2: Yes

5. Is the manuscript presented in an intelligible fashion and written in standard English?

Reviewer #1: Yes

Reviewer #2: No

6. Review Comments to the Author

Reviewer #1: Now, the paper is ready for publication except for a small issue: it is suggested to include one more row at the end of Table 7 mentioning "This Work" and then write the supporting results in subsequent columns. It will help the reader to understand the novelty of the proposed work at a glance. Congratulations on such a wonderful work. I request to extend such work in the future with some other methodologies.

Reviewer #2: First of all, I appreciate the authors for the perfect responses and improvements. I am delighted to inform you that this manuscript is very strong now, and I believe that this paper will gain a high number of citations in the future. The authors made a major correction from the previous round. The revised manuscript can be accepted for publication. I only have a minor notice, which can be considered in the final version. The paper is not required to be sent again to the reviewer. You just consider this note before publishing the paper. Kindly find the following note for further improvements:

1. In technical writing, it is not true to write “we, I, our, etc.” There are still some places that were not corrected, or some places in the new content were not written considering this comment.

7. PLOS authors have the option to publish the peer review history of their article (what does this mean?). If published, this will include your full peer review and any attached files.

Reviewer #1: **Yes: **DR. TARUN KUMAR DAS

Reviewer #2: No

---

## [Author Response · Author response to Decision Letter 2]

20 Jul 2025

For reviewer 1 : Done with thanks as in page 31 from line 733 to line 734.

For reviewer 2: Corrected with thanks as below:

- For (we), it has been corrected as in lines (285,315,423,425).

- For (our) it has been corrected as in line (106,215,360,371,728).

- For (I) it has been fixed as in line (850)

---

## [Editor Report · Decision Letter 2]

1 Aug 2025

An Adaptive Sliding Mode Controller with Free-Will Arbitrary Time Convergence for Three-Phase Rectifiers in Autonomous Agricultural Vehicles

PONE-D-25-20009R2

Dear Dr. Mahmood,

We’re pleased to inform you that your manuscript has been judged scientifically suitable for publication and will be formally accepted for publication once it meets all outstanding technical requirements.

Kind regards,

Akhtar Rasool, Ph.D.

Academic Editor

PLOS ONE

Additional Editor Comments (optional):

Congratulations
---

## [Editor Report · Acceptance letter]

PONE-D-25-20009R2

PLOS ONE

Dear Dr. Mahmood,

I'm pleased to inform you that your manuscript has been deemed suitable for publication in PLOS ONE. Congratulations! Your manuscript is now being handed over to our production team.

Kind regards,

on behalf of

Dr. Akhtar Rasool

Academic Editor

PLOS ONE